# Pre-COVID-19 pandemic health-related behaviours in children (2018–2020) and association with being tested for SARS-CoV-2 and testing positive for SARS-CoV-2 (2020–2021): a retrospective cohort study using survey data linked with routine health data in Wales, UK

Emily Marchant [1,2] Emily Lowthian [1,2] Tom Crick [2] Lucy J Griffiths [1] Richard Fry [1,3] Kevin Dadaczynski,[4,5] Orkan Okan,[6] Michaela James [1,3] Laura Cowley,[1] Fatemeh Torabi,[1] Jonathan Kennedy [1,3] Ashley Akbari [1] Ronan Lyons [1] Sinead Brophy [1,3]

For numbered affiliations see end of article.

**Correspondence to**
Dr Emily Marchant;
e.k.marchant@swansea.ac.uk

## ABSTRACT

**Objectives** Examine if pre-COVID-19 pandemic (prior March 2020) health-related behaviours during primary school are associated with (1) being tested for SARS-CoV-2 and (2) testing positive between 1 March 2020 and 31 August 2021.

**Design** Retrospective cohort study using an online cohort survey (January 2018 to February 2020) linked with routine PCR SARS-CoV-2 test results.

**Setting** Children attending primary schools in Wales (2018–2020), UK, who were part of the Health and Attainment of Pupils in a Primary Education Network (HAPPEN)_school network.

**Participants** Complete linked records of eligible participants were obtained for n=7062 individuals. 39.1% (n=2764) were tested (age 10.6±0.9; 48.9% girls) and 8.1% (n=569) tested positive for SARS-CoV-2 (age 10.6±1.0; 54.5% girls).

**Main outcome measures** Logistic regression of health-related behaviours and demographics were used to determine the ORs of factors associated with (1) being tested for SARS-CoV-2 and (2) testing positive for SARS-CoV-2.

**Results** Consuming sugary snacks (1–2 days/week OR=1.24, 95% CI 1.04 to 1.49; 5–6 days/week OR=1.31, 95% CI 1.07 to 1.61; reference 0 days), can swim 25 m (OR=1.21, 95% CI 1.06 to 1.39) and age (OR=1.25, 95% CI 1.16 to 1.35) were associated with an increased likelihood of being tested for SARS-CoV-2. Eating breakfast (OR=1.52, 95% CI 1.01 to 2.27), weekly physical activity ≥60 min (1–2 days OR=1.69, 95% CI 1.04 to 2.74; 3–4 days OR=1.76, 95% CI 1.10 to 2.82; reference 0 days), out-of-school club participation (OR=1.06, 95% CI 1.02 to 1.10), can ride a bike (OR=1.39, 95% CI 1.00 to 1.93), age (OR=1.16, 95% CI 1.05 to 1.28) and girls (OR=1.21, 95% CI 1.00 to 1.46) were associated with an

increased likelihood of testing positive for SARS-CoV-2. Living in least deprived areas (quintile 4 OR=0.64, 95% CI 0.46 to 0.90; quintile 5 OR=0.64, 95% CI 0.46 to 0.89) compared with the most deprived (quintile 1) was associated with a decreased likelihood.

**Conclusions** Associations may be related to parental health literacy and monitoring behaviours. Physically active behaviours may include coparticipation with others and exposure to SARS-CoV-2. A risk-versus-benefit approach must be considered in relation to promoting these health

## STRENGTHS AND LIMITATIONS OF THIS STUDY

⇒ Investigation of the association of prepandemic child health-related behaviour measures with subsequent SARS-CoV-2 testing and infection.

⇒ Reporting of multiple child health behaviours linked at an individual level to routine records of SARS-CoV-2 testing data through the Secure Anonymised Information Linkage Databank, using complete case analysis.

⇒ Child-reported health behaviours were measured before the COVID-19 pandemic (1 January 2018 to 28 February 2020) which may not reflect behaviours during COVID-19.

⇒ Health behaviours captured through the national-scale Health and Attainment of Pupils in a Primary Education Network (*HAPPEN*) survey represent children attending schools that engaged with the *HAPPEN* Wales primary school network and may not be representative of the whole population of Wales.

⇒ The period of study for PCR testing includes a time frame with varying prevalence rates, approaches to testing children (targeted and mass testing) and restrictions which were not measured in this study.

behaviours, given the importance of health-related behaviours such as childhood physical activity for development.

## BACKGROUND

The COVID-19 pandemic caused by SARS-CoV-2 has resulted in widespread disruption to the lives of children across the world, and has contributed to widened inequalities in children's health, well-being and education.[1 2] Childhood is a critical developmental period during which health behaviours are established which transcend into adolescence and adulthood.[3] The Organisation for Economic Co-operation and Development (OECD) recognised current trends in children's health, highlighting typical health behaviours of school-age children that warrant further research in order to better design policies that improve children's health outcomes.[4 5] These include nutrition-related behaviours such as fruit and vegetable intake, consumption of sugary foods and breakfast consumption, physical activity and sedentary behaviours and sleep. The establishment of these health behaviours during childhood is highly influenced by parental mechanisms and monitoring behaviours, particularly in children aged under 12.[6–8]

While evidence has demonstrated the negative impact of the COVID-19 pandemic on children's health-related behaviours including reduced physical activity, increased sedentary behaviour and poorer nutrition,[1 9] it is unclear if this association is bidirectional. That is, whether these health behaviours are associated with likelihood of SARS-CoV-2 infection. Within the adult population, emerging evidence suggests a plausible relationship between prepandemic health risk behaviours such as physical inactivity and poor nutrition with SARS-CoV-2 infection and severity of disease,[10–13] and increased risk of other infectious diseases.[14] This is attributed to the important role health behaviours play in shaping cardiometabolic health and immune system function. Indeed, research shows links to the early years including critical early developmental stages with subsequent risk of developing chronic inflammation, which is associated with non-communicable disease risk and mortality during adulthood.[15] Health behaviours such as adequate nutrient intake[16] and physical activity[17] are required for the regulation and function of the immune system.

As a result, researchers have advocated for consideration to be placed on the role of these health behaviours in future endemic/pandemic scenarios.[17] However, research to date has concentrated on adults, explored single health behaviours or examined those with severe COVID-19 infection and hospitalisation.[18 19] The focus of research within the childhood population has principally been placed on clinical outcomes as opposed to lifestyle outcomes, including identifying the clinical characteristics of severe infection, the presence of comorbidities, common symptoms such as cough and clinical biomarkers.[20 21] While serious COVID-19 illness in children is relatively rare, mild or asymptomatic infection

is common.[22] Positive SARS-CoV-2 tests require periods of self-isolation, impacting children's physical health and well-being, limiting opportunities for children to engage in health-promoting behaviours essential for optimal development such as regular physical activity.[9 23] Therefore, research examining the role of these health behaviours in a childhood population within the context of the COVID-19 pandemic is warranted.

Identifying the prepandemic health-related behavioural characteristics of children requiring a SARS-CoV-2 test or testing positive for SARS-CoV-2 infection and hypothesising potential mechanisms through which these may operate, including exposures, sociodemographic and parental influences, could yield insight to inform the current COVID-19 pandemic and future pandemic/endemic scenarios. This can also allow targeted intervention to minimise transmission risk that complements national public health measures and guidelines, and importantly, mitigates the disruption to children's lives, and prevent further exacerbation of pre-existing inequalities, safeguarding their health, well-being and education.

In Wales (one of the four nations of the UK, with devolved health and social care policies), approaches to performing PCR tests on children during the period of study included the presence of COVID-19 symptoms, if identified as a close contact to a positive case (eg, household contacts), or as a follow-up PCR test as encouraged in guidance at the time following a positive lateral flow test (LFT) (eg, showing symptoms or a close contact and having a positive LFT performed in the home).[24] Uptake of testing within the childhood population requires parental monitoring behaviours; for example, providing transport to testing facilities and parental health literacy through identification of symptoms.

This study investigates the association of prepandemic (prior to 1 March 2020) health-related behaviours self-reported by children aged 8–11 years during primary school before the COVID-19 pandemic between 1 January 2018 and 28 February 2020, with two outcomes: the odds of ever having a SARS-CoV-2 PCR test and the odds of ever testing positive for SARS-CoV-2 during the period of study. We aim to examine whether these self-reported markers of health-related behaviours reported before pandemic are associated with the likelihood of: (1) ever being tested for SARS-CoV-2 and (2) ever testing positive for SARS-CoV-2 between 1 March 2020 and 31 August 2021.

## METHODS

### Study design

This retrospective cohort study was conducted through the Health and Attainment of Pupils in a Primary Education Network (*HAPPEN*) primary school network.[25] *HAPPEN* was established in Wales, UK in 2014, following research with head teachers who advocated for increased collaboration to prioritise pupils' health and well-being,[26 27] and is a platform for conducting school-based

research.[2] [28–30] The network brings together primary schools with research and runs up to the current date. School participation in *HAPPEN* is voluntary and is either once, annually or biannually (eg, to evaluate school-based interventions). Through *HAPPEN*, children aged 8–11 (years 4–6) complete the *HAPPEN* survey, an online cohort survey that captures a range of validated self-reported health behaviours including physical activity, nutrition and sleep.[31] Retrospective health-related behaviour data were obtained from responses from the *HAPPEN* survey completed before pandemic between 1 January 2018 and 28 February 2020.

These retrospective survey responses were linked with PCR SARS-CoV-2 test results obtained from the Pathology COVID-19 Daily (PATD) routine data set between 1 March 2020 and 31 August 2021. The PATD data set contains pillar 1 (swab testing in Public Health England labs, National Health Service (NHS) Wales labs and NHS hospitals for those with a clinical need, and health and care workers) and pillar 2 (swab testing for the wider population, as set out in government guidance) individual results from PCR tests (negative (suspected), positive (confirmed) for SARS-CoV-2).[32] The period of interest (1 March 2020 to 31 August 2021) includes a time frame of varying approaches to testing children, documented in timeline format in online supplemental appendix 1.[32] This includes targeted (ie, symptomatic and suspected positive cases, identified as a close contact of a positive case) and mass testing (ie, between February 2021 and April 2021, the use of LFTs in the school setting for pupils aged 11 and above (secondary school age) to identify asymptomatic positive cases, with guidance for positive LFTs encouraging follow-up PCR tests).

Linkage was performed using the *Secure Anonymised Information Linkage* (SAIL) Databank.[33–35] Data were linked at the individual level using an anonymous linkage field (ALF) to identify participants and link SARS-CoV-2 test results (figure 1). The REporting of studies Conducted using Observational Routinely-collected Data (RECORD) checklist[36] for this study is presented in online supplemental appendix 2.

## Ethics
Electronic data (survey responses) were stored in secure files only accessible to the research team. The routine data used in this study are available in the SAIL Databank and are subject to review by an independent Information Governance Review Panel (IGRP), to ensure proper and appropriate use of SAIL data. Before any data can be accessed, approval must be received from the IGRP. When access has been approved, it is accessed through a privacy-protecting safe haven and remote access system referred to as the SAIL Gateway. SAIL has established an application process to be followed by anyone who would like to access data. This study has been approved by the SAIL IGRP (project reference: 0911).

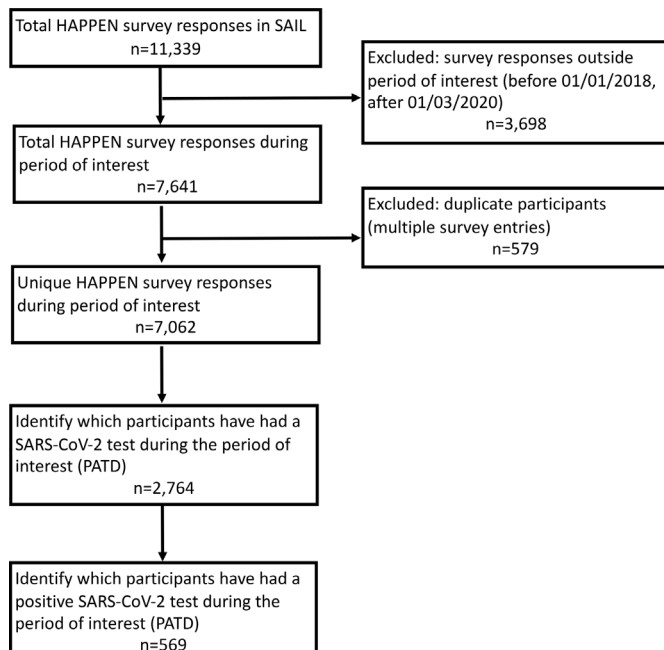

**Figure 1** Cohort flow diagram. HAPPEN, Health and Attainment of Pupils in a Primary Education Network; PATD, Pathology COVID-19 Daily; SAIL, Secure Anonymised Information Linkage.

## The *HAPPEN* survey and linked SAIL data
All primary schools (n=1203) in Wales, UK were invited to participate in the *HAPPEN* survey between 1 April 2014 and 28 February 2020 via a number of methods including email, social media promotion and through stakeholders in health and education (including local authority health and well-being teams, regional education consortia). Prior to 2018, *HAPPEN* was established in three of the local authorities (total n=22) in Wales. From 2018 to the period of interest, *HAPPEN* began its expansion to primary schools across Wales. Between 1 January 2018 and 28 February 2020, there were n=305 primary schools registered with *HAPPEN* (25% of primary schools in Wales). Participating in *HAPPEN* is voluntary and this study comprises a convenience sample of children attending n=129 primary schools (representing a 42% response rate of registered *HAPPEN* primary schools) from 16 out of 22 local authorities that participated in the *HAPPEN* survey during the period of interest (1 January 2018 and 28 February 2020). Schools were invited to share details of the survey with parents/guardians (including information sheets). To participate in the *HAPPEN* survey and link data to routine records, child assent was required in addition to parental consent (between 2014 and 2018) and opt-out parental consent (2019 onwards).

The *HAPPEN* survey is completed by children aged 8–11 as a self-guided activity within the school setting as a classroom activity with supervision from a teacher/teaching assistant. The survey takes approximately 30 min to complete and includes validated self-report measures of typical health behaviours including physical activity, screen time, nutrition, sleep and well-being.[31] A

full copy of the survey can be found in online supplemental appendix 3, and items, response categories and the coding framework included within analyses in online supplemental appendix 4.

The process of data coding involved two researchers. The first (MJ) cleaned the raw data (including checking for duplicate entries), removed identifiable information and generated a unique participant ID number to protect participants' anonymity. The second (EM) researcher coded the anonymised raw data set using STATA (V.16) to produce a data set for analyses. This *HAPPEN* data set was uploaded to the SAIL Databank, a trusted research environment containing individual-level anonymised population-scale data sources about the population of Wales, that enables secure data linkage and analysis for research, to be linked with SARS-CoV-2 testing data from the PATD data set. To link the data, the person-based identifiable data are separated from the survey data and sent to a trusted third party, Digital Health and Care Wales (the national organisation that designs and builds digital services for health and social care in Wales). The survey data are sent to SAIL using a secure file upload. A unique anonymous linkage field (ALF) is assigned to the person-based record before it is joined to clinical data via a system linking field. The ALF was used to link records at the individual level between the *HAPPEN* data set and PATD data set containing PCR testing data. This data set was accessible to authors listed from the Population Data Science group, Swansea University.

### Quantitative analysis

The primary outcomes were (1) whether the child was ever PCR tested for the SARS-CoV-2 virus and (2) whether the child had any positive SARS-CoV-2 test between 1 March 2020 and 31 August 2021. Participants were assigned a binary code for any SARS-CoV-2 test during the period of interest (1: PCR tested at least once for SARS-CoV-2 between 1 March 2020 and 31 August 2021; 0: no PCR SARS-CoV-2 test) and again for any positive SARS-CoV-2 test during the period of interest (1: any positive SARS-CoV-2 PCR test between 1 March 2020 and 31 August 2021; 0: negative PCR test for SARS-CoV-2; 0: not PCR tested for SARS-CoV-2 (unknown)). In the case of multiple PCR tests, the first occurrence was used. Participants were assumed to have remained in Wales during the period of interest. Eligibility criteria (see cohort flow diagram, figure 1) within final analyses models were any unique participant with complete linked survey and routine records. Inclusion dates of survey responses for analyses were between 1 January 2018 and 28 February 2020. Complete case multivariable logistic regression analyses, adjusting for confounding variables (sex, age on 1 March 2020, area-level deprivation using the Welsh Index of Multiple Deprivation (WIMD)[37] (version 2019)) and clustered by school (using sandwich estimator to account for differences between schools), determined the Odds Ratios (OR) for (1) ever being PCR tested for SARS-CoV-2 virus and (2) ever having a positive PCR

SARS-CoV-2 test during the period of interest. Missing categories of data (sex and WIMD data obtained through the SAIL Databank) were tested to see if they significantly predicted any outcomes.

Independent variables as measures of typical prepandemic health-related behaviours included within analyses were obtained retrospectively from the *HAPPEN* survey, completed between 1 January 2018 and 28 February 2020 (online supplemental appendix 4). Health-related behaviour measures included in multivariable analyses are recognised by the OECD as typical health behaviour trends during childhood that warrant research.[4 5] These related to the behaviours from the previous day (ate breakfast, travel actively to and/or from school, number of fruit/vegetable portions consumed, number of times teeth brushed, hours of sleep), frequency of behaviours every day, the previous 7 days (physically active ≥60 min, sedentary/screen time ≥2 hours, felt tired, ate a sugary snack) and general items including participation in number of out-of-school clubs, can ride a bike and can swim 25 m. A list of variables included in analyses, coding response categories and coding framework is presented in online supplemental appendix 4. Independent variables were entered concurrently and examined for association with the outcomes (1) ever PCR tested for SARS-CoV-2 and (2) ever tested positive for SARS-CoV-2 between 1 March 2020 and 31 August 2021.

### Patient and public involvement

The SAIL Databank has a Consumer Panel that provides the public's perspective on data linkage research. The Panel members are involved in all elements of the SAIL Databank process, from developing ideas, advising on bids through approval processes (via the independent IGRP), to disseminating research findings. For more information visit https://saildatabank.com/about-us/public-engagement/.

### RESULTS

Survey responses were obtained from n=11 339 participants (figure 1). Survey responses outside the period of interest (before 1 January 2018 and after 28 February 2020) were excluded (n=3698), followed by duplicate participants (occasions of multiple survey entries, n=579). In the case of duplicates, the most recent instance of survey participation was used. Complete linked unique records of participants meeting eligibility criteria were obtained for n=7062 individuals. Table 1 presents the descriptive statistics of the study sample by ever PCR tested for SARS-CoV-2 and ever tested positive for SARS-CoV-2 between 1 March 2020 and 31 August 2021. Of the total sample, 39.1% (n=2764) were PCR tested for SARS-CoV-2 and 8.1% (n=569) tested positive for SARS-CoV-2. The mean age on 1 March 2020 (start of period of interest) was 10.6 (±0.9) for those PCR tested (table 1) and 10.6 (±1.0) for those tested positive for SARS-CoV-2 (table 2). The time between the *HAPPEN* survey date and the SARS-CoV-2

**Table 1** Descriptive statistics of study sample by PCR tested for SARS-CoV-2 and PCR test positive for SARS-CoV-2 between 1 March 2020 and 31 August 2021

| | Tested for SARS-CoV-2 % (n) | Not tested for SARS-CoV-2 % (n) | Tested positive for SARS-CoV-2 % (n) | Tested negative/not tested (unknown) for SARS-CoV-2 % (n) |
|---|---|---|---|---|
| Sample | 39.1 (2764) | 60.9 (4298) | 8.1 (569) | 91.9 (6493) |
| Age at time of *HAPPEN* survey | 10.1±0.8 | 9.9±0.9 | 10.1±0.8 | 9.9±0.8 |
| Age on 1 March 2020 (start of period of interest) | 10.6±0.9 | 10.3±1.1 | 10.6±1.0 | 10.4±1.0 |
| Number of days between *HAPPEN* survey and SARS-CoV-2 test (median (IQR)) | 588 (385–685) | | 672 (599–715) | |
| Sex | | | | |
| Boy | 49.3 (1363) | 46.7 (2005) | 44.3 (252) | 48.0 (3116) |
| Girl | 48.9 (1352) | 51.8 (2226) | 54.5 (310) | 50.3 (3268) |
| Missing | 1.8 (49) | 1.5 (67) | 1.2 (7) | 1.7 (109) |
| WIMD 2019 quintiles | | | | |
| 1 (most deprived) | 24.3 (672) | 23.9 (1,025) | 28.5 (162) | 23.6 (1535) |
| 2 | 19.9 (551) | 19.02 (826) | 19.7 (112) | 19.5 (1265) |
| 3 | 16.5 (455) | 17.4 (748) | 17.6 (100) | 17.0 (1103) |
| 4 | 15.6 (431) | 15.8 (678) | 14.1 (80) | 15.9 (1029) |
| 5 (least deprived) | 18.0 (497) | 16.8 (771) | 16.5 (94) | 17.3 (1124) |
| Missing | 5.7 (158) | 7.0 (300) | 3.7 (21) | 6.7 (437) |

See online supplemental appendix 4 for variable codebook. Full descriptive statistics table is presented in online supplemental appendix 5.
HAPPEN, Health and Attainment of Pupils in a Primary Education Network; WIMD, Welsh Index of Multiple Deprivation.

PCR test date (median number of days (IQR)) was 588 (385–685) days for being PCR tested and 672 (599–715) days for PCR testing positive for SARS-CoV-2. Complete case analyses are presented. The maximum missing data were 7% (see table 1). We tested if missing categories of data (sex and WIMD obtained through the SAIL Databank) significantly predicted any outcomes and found that no missing categories significantly predicted the outcomes. Therefore, missing data were assumed to be at random through data linkage.[38] Unadjusted multivariable logistic regression analyses are presented in online supplemental appendix 5.

Table 2 presents the multivariable logistic regression for children ever PCR tested for SARS-CoV-2 between 1 March 2020 and 31 August 2021. The model showed a low goodness of fit ($R^2$=0.02). Children who reported to eat breakfast (OR=1.16, 95% CI 0.99 to 1.36, reference: did not eat breakfast, p<0.1), consume sugary snacks on 1–2 days (OR=1.24, 95% CI 1.04 to 1.49) and 5–6 days (OR=1.31, 95% CI 1.07 to 1.61) compared with 0 days, participate in more out-of-school clubs (OR=1.02, 95% CI 1.00 to 1.04), able to ride a bike (OR=1.15, 95% CI 0.98 to 11.35, reference: cannot ride a bike, p<0.1) and able to swim 25 m (OR=1.21, 95% CI 1.06 to 1.39, reference: cannot swim 25 m) were more likely to be PCR tested for SARS-CoV-2. Older children (OR=1.25, 95% CI 1.16 to 1.35) were also more likely to be PCR tested

for SARS-CoV-2, and compared with quintile 1 (most deprived), those in WIMD quintiles 3 (OR=0.85, 95% CI 0.70 to 1.03, p<0.1) and 5 (OR=0.85, 95% CI 0.72 to 1.02, p<0.1) were less likely to be PCR tested for SARS-CoV-2. Unadjusted multivariable logistic regression analyses are presented in online supplemental appendix 6.

Table 3 presents the multivariable logistic regression for children ever PCR tested positive for SARS-CoV-2 between 1 March 2020 and 31 August 2021. Children were more likely to test positive for SARS-CoV-2 if they reported to eat breakfast (OR=1.52, 95% CI 1.01 to 2.27, reference: did not eat breakfast), be physically active for ≥60 min on 1–2 days (OR=1.69, 95% CI 1.04 to 2.74), 3–4 days (OR=1.76, 95% CI 1.10 to 2.82) and 5–6 days (OR=1.59, 95% CI 0.93 to 2.73, p<0.1) compared with 0 days, participate in more out-of-school clubs (OR=1.06, 95% CI 1.02 to 1.10) and able to ride a bike (OR=1.39, 95% CI 1.00 to 1.93, reference: cannot ride a bike). Older children (OR=1.16, 95% CI 1.05 to 1.28) were more likely to test positive for SARS-CoV-2. Compared with boys, girls were more likely to test positive (OR=1.21, 95% CI 1.00 to 1.46), and compared with the most deprived (quintile 1), those living in the least deprived areas (quintile 4: OR=0.64, 95% CI 0.46 to 0.90; quintile 5: OR=0.64, 95% CI 0.46 to 0.89) were less likely to test positive for SARS-CoV-2. The model showed a low goodness of fit ($R^2$=0.02). Unadjusted multivariable logistic regression

**Table 2** Multivariable logistic regression model of significant health behaviour markers and probability of ever being PCR tested for SARS-CoV-2 between 1 March 2020 and 31 August 2021, accounting for baseline age, sex and deprivation, and clustered by school

| PCR tested for SARS-CoV-2 (n=6403, $R^2$=0.02) | OR | P value | 95% CI |
|---|---|---|---|
| Ate breakfast | 1.16* | 0.067 | 0.99 to 1.36 |
| *Reference: did not eat breakfast* | 1.00 | | |
| Actively travelled to school | 0.93 | 0.339 | 0.80 to 1.08 |
| *Reference: did not actively travel to school* | 1.00 | | |
| Actively travelled from school | 1.01 | 0.901 | 0.86 to 1.19 |
| *Reference: did not actively travel from school* | 1.00 | | |
| Number of fruit/vegetable portions | 1.00 | 0.959 | 0.97 to 1.03 |
| Number of times teeth brushed | 0.94 | 0.229 | 0.86 to 1.04 |
| Sleep hours | 1.01 | 0.682 | 0.97 to 1.04 |
| *Reference: 0 days physically active ≥60 min (previous 7 days)* | | | |
| 1–2 days physically active ≥60 min | 1.14 | 0.250 | 0.91 to 1.41 |
| 3–4 days physically active ≥60 min | 1.13 | 0.257 | 0.91 to 1.39 |
| 5–6 days physically active ≥60 min | 1.16 | 0.217 | 0.92 to 1.45 |
| 7 days physically active ≥60 min | 1.10 | 0.451 | 0.86 to 1.39 |
| *Reference: 0 days sedentary ≥2 hours (previous 7 days)* | 1.00 | | |
| 1–2 days sedentary ≥2 hours | 1.20 | 0.141 | 0.94 to 1.54 |
| 3–4 days sedentary ≥2 hours | 1.18 | 0.198 | 0.92 to 1.52 |
| 5–6 days sedentary ≥2 hours | 1.16 | 0.333 | 0.86 to 1.56 |
| 7 days sedentary ≥2 hours | 1.16 | 0.243 | 0.90 to 1.48 |
| *Reference: 0 days felt tired (previous 7 days)* | 1.00 | | |
| 1–2 days felt tired | 0.97 | 0.686 | 0.84 to 1.12 |
| 3–4 days felt tired | 1.00 | 0.963 | 0.85 to 1.16 |
| 5–6 days felt tired | 1.07 | 0.528 | 0.86 to 1.33 |
| 7 days felt tired | 0.97 | 0.728 | 0.83 to 1.14 |
| *Reference: 0 days consumed sugary snack (previous 7 days)* | 1.00 | | |
| 1–2 days consumed sugary snack | 1.24** | 0.018 | 1.04 to 1.49 |
| 3–4 days consumed sugary snack | 1.12 | 0.301 | 0.91 to 1.37 |
| 5–6 days consumed sugary snack | 1.31** | 0.008 | 1.07 to 1.61 |
| 7 days consumed sugary snack | 1.16 | 0.170 | 0.94 to 1.43 |
| Number of out-of-school club participations | 1.02* | 0.099 | 1.00 to 1.04 |
| Can ride a bike | 1.15* | 0.086 | 0.98 to 1.35 |
| *Reference: cannot ride a bike* | 1.00 | | |
| Can swim 25 m | 1.21** | 0.006 | 1.06 to 1.39 |
| *Reference: cannot swim 25 m* | 1.00 | | |
| Age on 1 March 2020 | 1.25** | < 0.001 | 1.16 to 1.35 |
| Sex (girl) | 0.92 | 0.161 | 0.81 to 1.04 |
| *Reference: sex (boy)* | 1.00 | | |
| *Reference: WIMD 2019 quintile 1 (Most deprived)* | 1.00 | | |
| WIMD 2019 quintile 2 | 0.95 | 0.600 | 0.80 to 1.14 |
| WIMD 2019 quintile 3 | 0.85* | 0.090 | 0.70 to 1.03 |
| WIMD 2019 quintile 4 | 0.87 | 0.131 | 0.73 to 1.04 |
| WIMD 2019 quintile 5 (Least deprived) | 0.85* | 0.078 | 0.72 to 1.02 |

*P<0.1; **p<0.05.
See online supplemental appendix 4 for variable codebook. Low to moderate correlation between variables (coefficients –0.19 to 0.71).
Complete case analysis.
WIMD, Welsh Index of Multiple Deprivation.

**Table 3** Multivariable logistic regression model of significant health behaviour markers and probability of ever PCR testing positive for SARS-CoV-2 between 1 March 2020 and 31 August 2021, accounting for baseline age, sex and deprivation, and clustered by school

| PCR test positive for SARS-CoV-2 (n=6403, $R^2$=0.02) | OR | P value | 95% CI |
|---|---|---|---|
| Ate breakfast _Reference: did not eat breakfast_ | 1.52** 1.00 | 0.043 | 1.01 to 2.27 |
| Actively travelled to school _Reference: did not actively travel to school_ | 0.91 1.00 | 0.481 | 0.70 to 1.18 |
| Actively travelled from school _Reference: did not actively travel from school_ | 0.98 1.00 | 0.910 | 0.72 to 1.33 |
| Number of fruit/vegetable portions | 0.98 | 0.461 | 0.94 to 1.03 |
| Number of times teeth brushed | 1.05 | 0.542 | 0.90 to 1.21 |
| Sleep hours | 0.97 | 0.345 | 0.92 to 1.03 |
| _Reference: 0 days physically active ≥60 min (previous 7 days)_ | 1.00 | | |
| 1–2 days physically active ≥60 min | 1.69** | 0.035 | 1.04 to 2.74 |
| 3–4 days physically active ≥60 min | 1.76** | 0.018 | 1.10 to 2.82 |
| 5–6 days physically active ≥60 min | 1.59* | 0.091 | 0.93 to 2.73 |
| 7 days physically active ≥60 min | 1.50 | 0.158 | 0.85 to 2.65 |
| _Reference: 0 days sedentary ≥2 hours (previous 7 days)_ | 1.00 | | |
| 1–2 days sedentary ≥2 hours | 0.96 | 0.847 | 0.63 to 1.47 |
| 3–4 days sedentary ≥2 hours | 0.94 | 0.789 | 0.59 to 1.50 |
| 5–6 days sedentary ≥2 hours | 0.93 | 0.803 | 0.51 to 1.68 |
| 7 days sedentary ≥2 hours | 1.02 | 0.946 | 0.63 to 1.65 |
| _Reference: 0 days felt tired (previous 7 days)_ | 1.00 | | |
| 1–2 days felt tired | 1.18 | 0.207 | 0.91 to 1.51 |
| 3–4 days felt tired | 1.17 | 0.232 | 0.91 to 1.50 |
| 5–6 days felt tired | 1.19 | 0.243 | 0.89 to 1.60 |
| 7 days felt tired | 0.89 | 0.390 | 0.68 to 1.16 |
| _Reference: 0 days consumed sugary snack (previous 7 days)_ | 1.00 | | |
| 1–2 days consumed sugary snack | 1.13 | 0.523 | 0.77 to 1.65 |
| 3–4 days consumed sugary snack | 1.06 | 0.783 | 0.70 to 1.61 |
| 5–6 days consumed sugary snack | 1.36 | 0.159 | 0.89 to 2.08 |
| 7 days consumed sugary snack | 1.08 | 0.727 | 0.71 to 1.63 |
| Number of out-of-school club participations | 1.06** | 0.002 | 1.02 to 1.10 |
| Can ride a bike _Reference: cannot ride a bike_ | 1.39** 1.00 | 0.049 | 1.00 to 1.93 |
| Can swim 25 m _Reference: cannot swim 25 m_ | 1.14 | 0.324 | 0.88 to 1.48 |
| Age on 1 March 2020 | 1.16** | 0.003 | 1.05 to 1.28 |
| Sex (girl) _Reference: sex (boy)_ | 1.21** 1.00 | 0.046 | 1.00 to 1.46 |
| Reference: WIMD 2019 quintile 1 (most deprived) | 1.00 | | |
| WIMD 2019 quintile 2 | 0.79 | 0.113 | 0.59 to 1.06 |
| WIMD 2019 quintile 3 | 0.79 | 0.128 | 0.59 to 1.07 |
| WIMD 2019 quintile 4 | 0.64** | 0.009 | 0.46 to 0.90 |
| WIMD 2019 quintile 5 | 0.64** | 0.008 | 0.46 to 0.89 |

*P<0.1; **p<0.05.
See online supplemental appendix 4 for variable codebook. Low to moderate correlation between variables (coefficients −0.19 to 0.71).
Complete case analysis.
WIMD, Welsh Index of Multiple Deprivation.

analyses are presented in online supplemental appendix 6.

## DISCUSSION

This study examines whether markers of health-related behaviours reported by primary school-age children between January 2018 and February 2020 are associated with the likelihood of ever being PCR tested for SARS-CoV-2 and ever testing positive between 1 March 2020 and 31 August 2021. Findings suggest that eating breakfast, weekly sugary snack consumption (both low and high), participating in more out-of-school clubs, being able to ride a bike and being able to swim 25 m were associated with an increased likelihood of being tested for SARS-CoV-2. Health behaviours associated with an increased likelihood of testing positive for SARS-CoV-2 were eating breakfast, engaging in higher weekly physical activity, participating in more out-of-school clubs and riding a bike. Boys were more likely to test positive for SARS-CoV-2 than girls, and those living in a less deprived area were less likely to test positive than those residing in the most deprived area.

This study encompasses a period of both targeted and mass PCR testing, and detecting positive child cases using routine PCR testing data in this study requires a parent/guardian to take the child for testing. We find associations between child-reported health-related behaviours with both PCR testing for SARS-CoV-2 and testing positive for SARS-CoV-2. Through this, we theorise that because health behaviours are largely guided and facilitated by parents, our associations are likely to be reflecting health literacy among parents, along with monitoring behaviours. In the case of symptomatic testing, the detection of positive child cases relies on parents recognising symptoms and communication with their child. For asymptomatic testing through the use of LFT (eg, asymptomatic school testing between February and April 2021), guidance encouraged positive LFTs to be followed up with PCR testing, requiring knowledge of how to access testing services and ability to access services (eg, transport). These behaviours form a level of health literacy, recognised as the ability to access, understand, interpret and apply medical information and make informed decisions regarding medical advice, issues or guidelines.[39] Parental health literacy impacts the decision a parent makes relating to their child[40] and is correlated with a number of health indicators including knowledge of health and health services, and the parent and child engaging in health-promoting behaviours.[8 39]

Parenting is an important contributor to promoting positive health behaviours in children, and is represented by a constellation of attitudes, behaviours and values for the child. The presence of multiple physically active behaviours represented by the association of being able to swim, ride a bike and participation in more out-of-school clubs may represent underlying parental involvement and modelling behaviour, including involvement in leisure-time activities, providing financial and transport provision to attend organised activities such as access to swimming lessons and the provision of equipment.[7] This may also have a socioeconomic component, building on the ideas of Bourdieu in terms of social capital, and accessing health-enhancing material items.[41]

Diet-related findings of eating breakfast and restrictive weekly sugary snack consumption (1–2 days/week) may indicate higher parental monitoring, supporting our theory. In comparison, higher weekly sugary snack consumption (5–6 days/week) may represent less restrictive parental monitoring and more autonomy and choice for the child. We posit that as parental behaviours are often driven by underlying styles of parenting,[42] the associations could be depicting varying levels of control; for instance, those snacking one to two times perhaps have parents with greater control versus those snacking five to six times with parents with less controlling styles. This theory may well transcend into other behaviours, including limits and freedom in socialising with others, placing a greater likelihood of infection of illness—including COVID-19.

While evidence recognises the importance of adequate nutrition[16] and physical activity[17] for cardiometabolic health and immune system function, the findings in the current study draw attention to another potential mechanism of increased contacts and exposure to SARS-CoV-2. Engagement in physically active behaviours such as out-of-school clubs, higher frequency of physically active days in a week and riding a bike may increase the number of social contacts of the child. Indeed, there is a wealth of evidence demonstrating that childhood physical activity participation is highly influenced by their social environment and coparticipation with peers.[43] It is therefore possible that physically active children had increased social contacts and exposure to SARS-CoV-2 through coparticipation of activity and play opportunities.

However, it is important to note that physical activity is an essential health behaviour required for optimal development and a range of health and well-being outcomes. These findings must be considered in balance with the importance of encouraging these behaviours and providing physically active opportunities during childhood. This viewpoint was also reflected in government guidance and risk assessments during the COVID-19 pandemic through the reopening of children's playgrounds and outdoor play spaces, with explicit reference to outdoor play and physical activity as fundamental for children's development and well-being.[44]

Contact patterns may also explain sex differences observed in this study, as we found girls are more likely to test positive for SARS-CoV-2. In addition to age assortative mixing patterns of children, there is a developmental tendency by children to socially interact with members of the same sex and engage in gender-type activity.[45] For girls, the location of play preferences is more likely to be indoors and in contact with supervising adults, where exposure to SARS-CoV-2 is possibly greater.[46] The findings of association between increasing age and likelihood

of testing positive for SARS-CoV-2 in this study are supported by wider literature which suggests increasing susceptibility of infection in the adolescent age group compared with younger than 10–14 years.[47]

Our findings also show an area-level social gradient. Those living in the least deprived areas (WIMD quintiles 4 and 5) were less likely to test positive for SARS-CoV-2 compared with the most deprived (quintile 1), which may reflect deprivation-related exposure patterns to SARS-CoV-2. Indeed, research conducted using the WIMD and English area-level deprivation indicators found adults living in the most deprived areas demonstrated differential exposures to SARS-CoV-2.[48] This included patterns of public activities such as attending work or education outside of the household, using public transport and car sharing with non-household members. This, and considerations of the deprivation-related disparities in the built environment including access to open spaces, highlights the inequalities that persist in SARS-CoV-2 infection. Furthermore, while it is likely that children mix with others from similar demographic areas, the finding in our study may also reflect community prevalence which was not captured.

## CONCLUSION

We theorise that health-promoting behaviours associated with a child being tested for SARS-CoV-2 and being identified as positive may be a proxy of higher parental health literacy and monitoring behaviours. Furthermore, coparticipation in physically active behaviours with peers may increase exposure to SARS-CoV-2. This must be considered from a risk-versus-benefit approach in relation to promoting these health behaviours, given the importance of health-related behaviours such as physical activity during childhood for development and wellbeing. This national-level case study using survey data linked with routine health data in Wales provides insight into these issues from a devolved policy-making context, with the potential for replicability and portability to other jurisdictions.

**Author affiliations**
[1]Population Data Science, Medical School, Swansea University, Swansea, UK
[2]Department of Education and Childhood Studies, Swansea University, Swansea, UK
[3]National Centre for Population Health and Wellbeing Research, Medical School, Swansea University, Swansea, UK
[4]Department of Nursing and Health Science, Fulda University of Applied Sciences, Fulda, Germany
[5]Center for Applied Health Sciences, Leuphana University of Lüneburg, Lüneburg, Germany
[6]Faculty of Sport and Health Sciences, Technical University of Munich, München, Germany

**Acknowledgements** The authors would like to thank all the participating primary schools and pupils who took part in this study. This work was supported by the National Centre for Population Health and Wellbeing Research through the *HAPPEN* network. This study makes use of anonymised data held in the Secure Anonymised Information Linkage (SAIL) Databank. We would like to acknowledge all the data providers who make anonymised data available for research. We would also like to thank Dr Annemarie Docherty and Dr Olivia Swann from The University of Edinburgh for providing informal peer review input to the final draft.

**Contributors** EM and SB conceptualised the study design. EM and MJ acquired the data, and EM and JK were responsible for data curation. EM performed the statistical analysis, undertook the initial interpretation of the data and wrote the initial draft. EL and SB contributed to the writing of the manuscript and provided statistical guidance. EL, JK, SB, LC and RL provided critical interpretation of the data. The manuscript was critically reviewed and edited by EL, TC, LJG, RF, KD, OO, MJ, LC, FT, JK, AA, RL and SB. SB provided supervision and TC and LJG provided mentorship. EM is the guarantor of the study. EM, EL, TC, LJG, RF, KD, OO, MJ, LC, FT, JK, AA, RL and SB approved the final manuscript and agreed to be accountable for all aspects of the work.

**Funding** The Economic and Social Research Council (ESRC) funded the development of the *HAPPEN* network (grant number: ES/J500197/1) which this research was conducted through. The National Centre for Population Health and Wellbeing Research (NCPHWR) funded by Health and Care Research Wales provided infrastructural support for this work. This work was supported by the Con-COV team funded by the Medical Research Council (grant number: MR/V028367/1). This work was supported by Health Data Research UK, which receives its funding from HDR UK (HDR-9006) funded by the UK Medical Research Council, Engineering and Physical Sciences Research Council, Economic and Social Research Council, Department of Health and Social Care (England), Chief Scientist Office of the Scottish Government Health and Social Care Directorates, Health and Social Care Research and Development Division (Welsh Government), Public Health Agency (Northern Ireland), British Heart Foundation (BHF) and the Wellcome Trust. This work was a collaboration with the ADR Wales programme of work. ADR Wales is part of the Economic and Social Research Council (part of UK Research and Innovation) funded ADR UK (grant number: ES/S007393/1). This work was supported by the Wales COVID-19 Evidence Centre, funded by Health and Care Research Wales and by the COVID-19 Longitudinal Health and Wellbeing National Core Study funded by the Medical Research Council (MC_PC_20030).

**Competing interests** None declared.

**Patient and public involvement** Patients and/or the public were involved in the design, or conduct, or reporting, or dissemination plans of this research. Refer to the Methods section for further details.

**Patient consent for publication** Not applicable.

**Ethics approval** This study involves human participants and was approved by the Swansea University Medical School Research Ethics Committee (reference ID: 2017-0033H). Participants gave informed consent to participate in the study before taking part.

**Provenance and peer review** Not commissioned; externally peer reviewed.

**Data availability statement** Data are available upon reasonable request. The routine data used in this study are available in the SAIL Databank at Swansea University, Swansea, UK. All proposals to use SAIL data are subject to review by an IGRP. Before any data can be accessed, approval must be given by the IGRP. The IGRP gives careful consideration to each project to ensure proper and appropriate use of SAIL data. When access has been approved, it is gained through a privacy-protecting safe haven and remote access system referred to as the SAIL Gateway. SAIL has established an application process to be followed by anyone who would like to access data via SAIL: https://www.saildatabank.com/application-process. This study has been approved by the IGRP as project 0911.

**ORCID iDs**
Emily Marchant http://orcid.org/0000-0002-9701-5991
Emily Lowthian http://orcid.org/0000-0001-9362-0046
Tom Crick http://orcid.org/0000-0001-5196-9389
Lucy J Griffiths http://orcid.org/0000-0001-9230-624X
Richard Fry http://orcid.org/0000-0002-7968-6679
Michaela James http://orcid.org/0000-0001-7047-0049
Jonathan Kennedy http://orcid.org/0000-0002-1122-6502
Ashley Akbari http://orcid.org/0000-0003-0814-0801
Ronan Lyons http://orcid.org/0000-0001-5225-000X
Sinead Brophy http://orcid.org/0000-0001-7417-2858

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
