## [Reviewer comments · BMJ Open]

ARTICLE DETAILS

TITLE (PROVISIONAL)	Pre-COVID-19 pandemic health-related behaviours in children (2018-2020) and association with being tested for SARS-CoV-2 and testing positive for SARS-CoV-2 (2020-2021): a retrospective cohort study using survey data linked with routine health data in Wales, UK
AUTHORS	Marchant, Emily; Lowthian, Emily; Crick, Tom; Griffiths, Lucy; Fry, Richard; Dadaczynski, Kevin; Okan, Orkan; James, M; Cowley, Laura; Torabi, Fatemeh; Kennedy, Jonathan; Akbari, Ashley; Lyons, Ronan; Brophy, Sinead

VERSION 1 – REVIEW

REVIEWER	Anderson , Laura McMaster University Centre for Health Economics and Policy Analysis, Health Research Methods, Evidence and Impact
REVIEW RETURNED	23-Feb-2022

GENERAL COMMENTS	Thank you for the opportunity to review this paper. This is a retrospective cohort study of nearly 7000 children evaluating the association between pre-pandemic health behaviours and covid-19 testing. I think it is a valuable contribution and suggest some revisions to clarify the study. 1. Based on the title I expected the study results to be about how pre-pandemic behaviours in children were associated with risk of COVID-19 (i.e., SARS-Cov2 infection) but the results seem to suggest that this manuscript is actually about whether pre-pandemic behaviours were associated with ever being tested for COVID. I suggest that this be clarified in the title and throughout the abstract.2. The outcome variables for the two sentences in the results section of the abstract are unclear. What is the differences between “being tested for SARS-COV-2” and “increased likelihood of testing”. Is the latter actually supposed to say “increased likelihood of testing positive”?3. Please indicate the comparator group throughout whenever presenting an odds ratio.4. What do you mean by “profiling research” line 13, page 6?5. Line 25-31 page 6, these details about the vaccine roll-out are likely no longer accurate and not really relevant. Consider removing.6. Could you change the word “prior” throughout the manuscript and title to “pre-pandemic” or a more meaningful time period description?7. Could you clarify the description of when children were eligible for testing? It is unclear to me if the focus of this manuscript it about access to testing (e.g. unmet healthcare needs) or if testing
---

	was widely available and it was about testing positive. Were there any circumstances in which children needed to provide negative tests to participate in extracurricular activities or travel (i.e., was a test only when symptomatic or also to show health status)? 8. Please state the study design. I think this was a retrospective cohort study but it is not stated in the methods. 9. What was the response rate for HAPPEN? How were the schools selected? 10. How did you know if all children remained in the outcome dataset? Could you discard those who moved or did you assume they just had no test? 11. How did you adjust for clustering by school in logistic regression? And, did you use repeated measures of exposure and/or outcome per child? If so, how did you adjust for those repeated measures? If not, please explain how the unique exposure and outcome time points were selected. 12. Why were all predictor variables collapsed into binary variables? 13. Could you clarify the model building approach? It first says each model was adjusted for a few a priori identified confounders but then later in the methods it says a manual step-wise approach was used. These seem to contradict each other (e.g., causal modeling vs prediction). It would be helpful if a strong rationale and references were provided to support the model building strategy. 14. Consider using past tense throughout to describe the results and conclusions. 15. I don't understand the conclusion in the discussion that says "this study did not find evidence that reporting positive health-related behaviours is associated with a reduced odds of being tested or testing positive for SARS-CoV-2." It might be helpful to clarify the hypothesized direction of associations in the background. 16. Could you discuss whether you think the timing of the exposure measurement is a relevant time period? Would you expect sleep duration (or any of the exposures) up to 6 years before the pandemic to be associated with COVID-19 risk? 17. A few places in the manuscript it is mentioned that the clustering of health behaviours is important, but it doesn't appear that the clustering of behaviours was examined. Was it?
--	--

REVIEWER	Cosma, Alina Trinity College Dublin, Sociology
REVIEW RETURNED	13-Mar-2022

GENERAL COMMENTS	Dear authors The manuscript entitled 'Prior health-related behaviours in children (2014-2020) and association with a positive SARS-CoV-2 test during adolescence (2020-2021): a retrospective cohort study using survey data linked with routine health data in Wales, UK' aims to examine if prior health-related behaviours during primary school are associated with being tested for SARS-CoV-2 and testing positive during adolescence. The paper has the potential to contribute significantly to the body of literature, however due to the concerns outlined below I cannot recommend its publication. First and foremost, I would invite the authors to take a step back and consider the evidence and theoretical underpinnings of their investigation (ie, research questions). The literature mentioned in
---

the Introduction it is rather generic and does not support directly the research questions. The authors argue that this is the first studies to investigate the associations between previous health behaviours and subsequent (positive) SARS-CoV-2 test during adolescence, however no proper explanation is provided for why would one expect such associations. The references #8 to#11 mentioned in the introduction tangentially address this, and the references #12& #13 might not be enough to justify this approach.

Secondly, the authors do not prepare at all the reader with regards to which 'health behaviours' are being investigated. As this term is rather generic and as it transpires later in the Methods section, the options of health behaviours is rather small and unusual to use as a predictor for research questions as these.

The Methods section is well written and enough information about the initial data collection and data linkage is presented. Whereas there are no issues there, I would like to return to the first point I made above and ask the researchers to reflect on why would they expect to see associations between very specific questions that ask about frequency of health behaviours in the day (eg., eating breakfast), week before (e.g., PA) or usually (e.g., can ride a bike) and (sometime) 6years later SARS-COV-2 infection?

Although this could be seen as an exploratory investigation, both models have an extremely poor goodness-of-fit (explained variance) (i.e., $R^2=0.006$, respectively $R^2=0.02$). The authors mentioned this aspect briefly in the results but do not expand on this in their discussion. Moving beyond the statistical significance of their results (ie, identified predictors) what is the ecological significance of these results given that over 99% of model is explained by other factors? I would be reluctant to imply any mechanisms on models with such poor fit.

The results show that positive health behaviours such as reporting the recommended level of sleep (at least nine hours), participating in at least three out of school clubs, being able to ride a bike and being able to swim 25 metres were associated with an increased likelihood of being tested for SARS-CoV-2. The authors hypothesises that this could be due to the parental involvement and health literacy but fail to suggest/consider any alternative explanations.

As a general observation the Introduction and Discussion are completely disconnected with each other. The former focuses extensively on parental health literacy aspect which is not investigated in this study and that is only suggested as a possible mechanism behind some poorly explaining regression models. Also the authors introduce more confusion in the Discussion by bringing in parental vaccination decisions and linking it with their suggested explanations for the findings. I would advise against adding these speculations.

VERSION 1 – AUTHOR RESPONSE

Reviewer 1

1) Based on the title I expected the study results to be about how pre-pandemic behaviours in children were associated with risk of COVID-19 (i.e., SARS-CoV2 infection) but the results seem to suggest that this manuscript is actually about whether pre-pandemic behaviours were associated with ever being tested for COVID. I suggest that this be clarified in the title and throughout the abstract.

Thank you for raising this. This study examines the association of pre-pandemic behaviours with two outcomes

- i) ever being tested for SARS-CoV-2
- ii) ever testing positive for SARS-CoV-2 between 1 March 2020 and 31 August 2021.

We have amended this in the title, abstract and throughout the manuscript to ensure there is clarity regarding the focus of this study on two outcomes:

Title: Pre-COVID-19 pandemic health-related behaviours in children (2018-2020) and association with being tested for SARS-CoV-2 and testing positive for SARS-CoV-2 (2020-2021): a retrospective cohort study using survey data linked with routine health data in Wales, UK (page 1)

Abstract: Objective: Examine if pre-COVID-19 pandemic (prior March 2020) health-related behaviours during primary school are associated with i) being tested for SARS-CoV-2 and ii) testing positive between 1 March 2020 to 31 August 2021.

Main outcome measures: Logistic regression of health-related behaviours and demographics were used to determine Odds Ratios (OR) of factors associated with i) being tested for SARS-CoV-2 and ii) testing positive for SARS-CoV-2. (page 2-3)

Background: We aim to examine whether these self-reported markers of health-related behaviours reported pre-pandemic are associated with the likelihood of; i) ever being tested for SARS-CoV-2 and ii) ever testing positive for SARS-CoV-2 between 1 March 2020 and 31 August 2021. (page 7 line 145-149)

Methods: Quantitative analysis: The primary outcomes were i) whether the child was PCR tested for the SARS-CoV-2 virus and ii) whether the child had a positive SARS-CoV-2 test between 1 March 2020 and 31 August 2021 (page 11 line 257-259)

Discussion: This study examines whether markers of health-related behaviours reported by children during primary school between January 2018 and February 2020 are associated with the likelihood of being PCR-tested for SARS-CoV-2 and testing positive between 1 March 2020 and 31 August 2021 (page 18 line 513-516)

We have also included additional sentences in the results to ensure clarity between tables for PCR-tested and testing positive for SARS-CoV-2:

Results: Table 2 presents the multivariable logistic regression for children ever PCR-tested for SARS-CoV-2 between 1 March 2020 and 31 August 2021. (page 14 line 422-423)

Table 3 presents the multivariable logistic regression for children ever PCR-tested positive for SARS-CoV-2 between 1 March 2020 and 31 August 2021. (page 16 line 462-463)

2) The outcome variables for the two sentences in the results section of the abstract are unclear. What is the differences between “being tested for SARS-COV-2” and “increased likelihood of testing”. Is the latter actually supposed to say “increased likelihood of testing positive”?

Thank you for raising this. This was a mistype in the original submission which we contacted the Editorial Office to amend. We have also updated the abstract to reflect the results from the revised manuscript, as below:

Abstract: Results: Consuming sugary snacks (1-2 days/week OR=1.24, 95% CI 1.04 - 1.49; 5-6 days/week 1.31, 1.07 - 1.61; reference 0 days) can swim 25m (1.21, 1.06 - 1.39) and age (1.25, 1.16 - 1.35) were associated with an increased likelihood of being tested for SARS-CoV-2. Eating breakfast (1.52, 1.01 - 2.27), weekly physical activity \geq 60 mins (1-2 days 1.69, 1.04 - 2.74; 3-4 days 1.76, 1.10 - 2.82, reference 0 days), out of school club participation (1.06, 1.02 - 1.10), can ride a bike (1.39, 1.00 - 1.93), age (1.16, 1.05 - 1.28) and girls (1.21, 1.00 - 1.46) were associated with an increased likelihood of testing positive for SARS-CoV-2 (1.16, 1.10 - 1.22). (page 3)

3) Please indicate the comparator group throughout whenever presenting an odds ratio.

We appreciate you drawing this to our attention and agree including the comparator reference group when presenting Odds Ratios is useful for the reader. We have amended this throughout the logistic regression results tables 2 and 3, with an example below:

PCR-tested for SARS-CoV-2
(n=6,403, R²=0.02) OR p 95% CI
Ate breakfast 1.16* 0.067 0.99 - 1.36
Reference: did not eat breakfast 1.00
Active travel to school 0.93 0.339 0.80 - 1.08
Reference: did not active travel to school 1.00
(page 14-17)

We have also indicated the reference group in the main body of the text when referring to Odds Ratios, for example:

Results: Children were more likely to test positive for SARS-CoV-2 if reporting to eat breakfast (OR=1.52, 95% CI 1.01 - 2.27, reference: did not eat breakfast), be physically active for \geq 60 mins on 1-2 days (1.69, 1.04 - 2.74), 3-4 days (1.76, 1.10 - 2.82) and 5-6 days (1.59, 0.93 - 2.73, p<0.1) compared to 0 days, participate in more out of school clubs (1.06, 1.02 - 1.10) and able to ride a bike (1.39, 1.00 - 1.93, reference: cannot ride a bike) (page 14-17)

4) What do you mean by “profiling research” line 13, page 6

We have amended this by removing the term “profiling research” as we agree that it was not clear for the reader, and provided further information:

Background: The focus of research within the childhood population has principally been placed on clinical outcomes as opposed to lifestyle outcomes, including identifying the clinical characteristics of severe infection, the presence of comorbidities, common symptoms such as cough and clinical

biomarkers[20,21]. (page 6 line 32-35)

5) Line 25-31 page 6, these details about the vaccine roll-out are likely no longer accurate and not really relevant. Consider removing.

Noted, this has been removed from the manuscript as we agree it is no longer relevant nor accurate within the rapidly changing pandemic.

6) Could you change the word “prior” throughout the manuscript and title to “pre-pandemic” or a more meaningful time period description?

Thank you for raising this important point. We have addressed this throughout the manuscript by replacing “prior” with “pre-pandemic”, and have included the meaning of pre-pandemic in the Abstract and Background:

Abstract: Objective: Examine if pre-COVID-19 pandemic (prior March 2020) health-related behaviours during primary school are associated with i) being tested for SARS-CoV-2 and ii) testing positive between 1 March 2020 to 31 August 2021. (page 2)

Background: This study investigates the association of pre-pandemic (prior to 1 March 2020) health-related behaviours self-reported by children aged 8-11 years during primary school before the COVID-19 pandemic (page 7 line 141-143)

7) Could you clarify the description of when children were eligible for testing? It is unclear to me if the focus of this manuscript is about access to testing (e.g. unmet healthcare needs) or if testing was widely available and it was about testing positive. Were there any circumstances in which children needed to provide negative tests to participate in extracurricular activities or travel (i.e., was a test only when symptomatic or also to show health status)?

Thank you for raising this important point regarding eligibility of testing for children. We agree that this requires further information in the manuscript.

We have included an additional reference from the Welsh Government research department documenting the timeline of testing approaches [24]:

[24] Senedd Research. Coronavirus timeline: Welsh and UK governments' response – IN BRIEF. 2020. <https://seneddresearch.blog/2020/03/19/covid-19-timeline-welsh-and-uk-governments-response/> (accessed 30 Mar 2022).

In addition, to aid interpretation for the reader, we have used this Welsh Government document to produce a new online supplemental appendix 1, following the timeline from reference 24 to present the timeline format of notable changes relevant to testing children and school operation during the period of investigation.

This appendix item presents changes in testing availability, and includes the use of Lateral Flow Tests (LFTs) in mass community and school (e.g. asymptomatic) testing approaches. Guidance at the time encouraged positive LFTs to be followed up with PCR tests, however it is possible that positive LFTs were not followed by PCR tests and these cases are undetected. We believe this to operate through the mechanisms of parental involvement and health literacy suggested in the manuscript. We have included an additional sentence in the manuscript to reflect this:

Background: In the case of symptomatic testing, the detection of positive child cases relies on parents recognising symptoms and communication with their child. For asymptomatic testing through the use of LFT (e.g. asymptomatic school testing between February and April 2021), guidance encouraged positive LFTs to be followed up with PCR-testing, requiring knowledge of how to access testing services and ability to access services (e.g. transport). (page 18-19 line 531-589)

We have also further clarified this in the manuscript, please see below:

Background: In Wales, one of the four nations of the UK, approaches to performing PCR tests on children during the period of study included the presence of COVID-19 symptoms, if identified as a close contact to a positive case (e.g. household contacts), or as a follow-up PCR test as encouraged in guidance at the time following a positive lateral flow test (e.g. showing symptoms or a close contact and having a positive lateral flow test performed in the home)[24]. (page 7 line 133-140)

Methods: Study design: The PATD dataset contains pillar 1 (swab testing in Public Health England (PHE) labs, NHS Wales labs and NHS hospitals for those with a clinical need, and health and care workers) and pillar 2 (swab testing for the wider population, as set out in government guidance) individual results from PCR tests (negative (suspected), positive (confirmed) for SARS-CoV-2 [29]. The period of interest (1 March 2020 to 31 August 2021) includes a time frame of varying approaches to testing children, documented in timeline format in online supplemental appendix 1[29]. This includes targeted (i.e. symptomatic and suspected positive case, identified as a close contact of a positive case) and mass testing (i.e. between February 2021 and April 2021 the use of Lateral Flow Tests (LFTs) in the school setting for pupils aged 11 and above (secondary school age) to identify asymptomatic positive cases, with guidance for positive LFTs encouraging follow up PCR tests). (page 8 line 180-190)

Strengths and limitations: The period of study for PCR-testing for and testing positive for SARS-CoV-2 includes a time frame with varying prevalence rates, approaches to testing children (targeted and mass testing) and restrictions which were not measured in this study. (page 3-4)

Regarding the possibility of children requiring negative tests to participate in extracurricular activities or travel, children were exempt from following 'COVID Pass' requirements of negative tests. See: <https://gov.wales/nhs-covid-pass-help-getting-your-pass>.

8) Please state the study design. I think this was a retrospective cohort study but it is not stated in the methods.

Thank you drawing our attention to this, this has been amended in the Methods section:

Methods: Study design: This retrospective cohort study was conducted through the HAPPEN primary school network (Health and Attainment of Pupils in a Primary Education Network)[25] (page 7 line 152-153)

9) What was the response rate for HAPPEN? How were the schools selected?

Thank you for raising this point. We have amended this in-text to provide the reader with information regarding the number of primary schools and local authorities, and that the study comprises of a convenience sample of primary schools that decided to participate in HAPPEN. Although all primary schools in Wales have been contacted (e.g. through direct email), the response rate is unclear as we

cannot confirm that emails reached the intended school, headteacher or health and wellbeing coordinator.

Methods: The HAPPEN survey and linked SAIL data: All primary schools (n=1,203) in Wales, UK were invited to participate in the HAPPEN survey between 1 April 2014 and 28 February 2020 via a number of methods including email, social media promotion and through stakeholders in health and education (including local authority health and wellbeing teams, regional education consortia). Prior to 2018, HAPPEN was established in three of the local authorities (total n=22) in Wales. From 2018 to the period of interest, HAPPEN began its expansion to primary schools across Wales. Participating in HAPPEN is voluntary and this study comprises of a convenience sample of children attending n=129 primary schools from 16 local authorities that participated in the HAPPEN survey during the period of interest (1 January 2018 and 28 February 2020). (page 9-10 line 213-225)

10) How did you know if all children remained in the outcome dataset? Could you discard those who moved or did you assume they just had no test?

We used complete case analyses and assumed no evidence of a PCR test as not being tested and not being positive, therefore coding this as a 0. We have further clarified this in the manuscript as below:

Methods: Quantitative analysis: Participants were assigned a binary code for SARS-CoV-2 test during period of interest (1: tested at least once for SARS-CoV-2 between 1 March 2020 and 31 August 2021, 0: no evidence of PCR SARS-CoV-2 test) and again for a positive SARS-CoV-2 test during period of interest (1; testing positive for SARS-CoV-2 between 1 March 2020 and 31 August 2021, 0; testing negative, 0; no evidence of PCR test). Participants were assumed to have remained in Wales during the period of interest. (page 11 line 259-264)

Furthermore, we have now refined the period of interest for health behaviour survey responses obtained through the HAPPEN survey from 2014 – 2020, to 2018 to 2020, and thus have assumed participants to have remained in Wales during the period of interest, as now stated in the manuscript extract above.

11) How did you adjust for clustering by school in logistic regression? And, did you use repeated measures of exposure and/or outcome per child? If so, how did you adjust for those repeated measures? If not, please explain how the unique exposure and outcome time points were selected.

Regarding clustering by school, a unique school code was assigned to school name obtained through the survey. Using STATA, the multivariable regression models were clustered by school using code: cluster(schoolcode).

Results: Quantitative analysis: Multivariable logistic regression analyses, adjusting for confounding variables (sex, age on 1 March 2020, area-level deprivation using the Welsh Index of Multiple Deprivation (WIMD)[34] (version 2019) and clustered by school (to account for differences between schools) (page 11 line 267-270)

Relating to your point about repeated measures – we used single measures for exposures. Any SARS-CoV-2 test and any positive SARS-CoV-2 test was flagged within the period of interest between 1 March 2020 and 31 August 2021 and participants assigned a binary code of 1==any test/positive test during period of interest, 0==no test/negative test during period of interest. This has been further clarified in the manuscript:

Results: Quantitative analysis: Participants were assigned a binary code for SARS-CoV-2 test during

period of interest (1: tested at least once for SARS-CoV-2 between 1 March 2020 and 31 August 2021, 0: no evidence of PCR SARS-CoV-2 test) and again for a positive SARS-CoV-2 test during period of interest (1; testing positive for SARS-CoV-2 between 1 March 2020 and 31 August 2021, 0; testing negative, 0; no evidence of PCR test). Participants were assumed to have remained in Wales during the period of interest. (page 11 line 259-264)

12) Why were all predictor variables collapsed into binary variables?

We originally wanted to estimate the associations between health behaviour guidelines and assign 'positive' or 'negative' codes to, e.g., physical activity every day, and SARS-2-CoV infection or testing. However, after consulting with EL who provided statistical guidance, we have decided to revert to more detailed measures e.g., number of hours slept, to retain power and nuance in our estimates. Tables 2 and 3 (pages 15-17) now present multivariable analyses including detailed response items reflective of survey response categories. For example:

PCR-tested for SARS-CoV-2

(n=6,403, R²=0.02) OR p 95% CI

Reference: 0 days physically active \geq 60 mins (previous seven days)

1-2 days physically active \geq 60 mins 1.14 0.250 0.91 - 1.41

3-4 days physically active \geq 60 mins 1.13 0.257 0.91 - 1.39

5-6 days physically active \geq 60 mins 1.16 0.217 0.92 - 1.45

7 days physically active \geq 60 mins 1.10 0.451 0.86 - 1.39

The above depicts full response options from corresponding survey item (presented in online supplemental appendix 3):

13) Could you clarify the model building approach? It first says each model was adjusted for a few a priori identified confounders but then later in the methods it says a manual step-wise approach was used. These seem to contradict each other (e.g., causal modeling vs prediction). It would be helpful if a strong rationale and references were provided to support the model building strategy.

Similarly to before, we consulted with EL who suggested that an exploratory theory-based model may provide better insight. From this, we have changed our step-wise approach and explored all areas of health behaviours. We recognise that our previous modelling strategy may have been weaker given our unclear rationale. In our revisions, we examine all health behaviours in a univariate model, then all together in a multivariate model, and then we adjust our multivariate model for confounders.

The health behaviours explored within analyses are recognised by the OECD as current trends in children's health behaviours, refer to reference [4-5], and we have clarified this in the background and methods as presented below.

Background: The Organisation for Economic Co-operation and Development (OECD) recognised current trends in children's health, highlighting typical health behaviours of school-aged children that warrant further research in order to better design policies that improve children's health outcomes[4,5]. These include nutrition-related behaviours such as fruit and vegetable intake, consumption of sugary foods and breakfast consumption, physical activity and sedentary behaviours and sleep. (page 5 line 7-12)

Methods: Quantitative analysis: Health-related behaviour measures included in multivariable analyses are recognised by the OECD as typical health behaviour trends during childhood[4,5]. (page 12 line

287-289)

[4] OECD. 'Are children active and physically healthy?'. In: Measuring What Matters for Child Well-Being and Policies. Paris: : OECD Publishing 2021. 1–295. doi:10.1787/e82fded1-en

[5] OECD. Education in the Digital Age. In: Burns T, Gottschalk F, eds. Education in the Digital Age. OECD 2020. 1–218. doi:10.1787/1209166A-EN

Furthermore, previous research demonstrates the importance of these health behaviours for cardiometabolic health and immune system function, and academics have advocated for greater importance to be placed on these health behaviours in future pandemics. This supports the exploratory theory-based model as suggested by EL. We have provided additional literature to support our stance within the manuscript:

Background: Within the adult population, emerging evidence suggests a plausible relationship between pre-pandemic health risk behaviours such as physical inactivity and poor nutrition with SARS-CoV-2 infection and severity of disease[10–13], and increased risk of other infectious diseases[14]. This is attributed to the role health behaviours play in shaping cardiometabolic health and immune system function. Indeed, research shows links to the early years including critical early developmental stages with subsequent risk of developing chronic inflammation, which is associated with non-communicable disease risk and mortality during adulthood [15]. Health behaviours such as adequate nutrient intake[16] and physical activity[17] are required for the regulation and function of the immune system.

As a result, academics have advocated for consideration to be placed on the role of these health behaviours in future endemic/pandemic scenarios[17]. However, research to date has concentrated on adults, explored single health behaviours or examined those with severe COVID-19 infection and hospitalisation[18,19]. The focus of research within the childhood population has principally been placed on clinical outcomes as opposed to lifestyle outcomes, including identifying the clinical characteristics of severe infection, the presence of comorbidities, common symptoms such as cough and clinical biomarkers[20,21]. Whilst serious COVID-19 illness in children is relatively rare, mild or asymptomatic infection is common[22]. Positive SARS-CoV-2 tests require periods of self-isolation, impacting children's physical health and wellbeing, limiting opportunities for children to engage in health-promoting behaviours essential for optimal development such as regular physical activity[9,23]. Therefore, research examining the role of these health behaviours in a childhood population within the context of the COVID-19 pandemic is warranted. (page 5-6 line 18-41)

Methods:

14) Consider using past tense throughout to describe the results and conclusions.

Thank you for drawing our attention to this, we have amended the results and conclusion as past tense, for example:

Results: Children were more likely to test positive for SARS-CoV-2 if reporting to eat breakfast (OR=1.52, 95% CI 1.01 - 2.27, reference: did not eat breakfast), be physically active for ≥ 60 mins on 1-2 days (1.69, 1.04 - 2.74), 3-4 days (1.76, 1.10 - 2.82) and 5-6 days (1.59, 0.93 - 2.73, $p < 0.1$) compared to 0 days, participate in more out of school clubs (1.06, 1.02 - 1.10) and able to ride a bike (1.39, 1.00 - 1.93, reference: cannot ride a bike).

15) I don't understand the conclusion in the discussion that says " this study did not find evidence that reporting positive health-related behaviours is associated with a reduced odds of being tested or testing positive for SARS-CoV-2." It might be helpful to clarify the hypothesized direction of

associations in the background.

Thank you for highlighting this, we have since updated the discussion and conclusion to ensure clarify for the reader, and have removed this sentence:

Discussion: This study examines whether markers of health-related behaviours reported by primary school-aged children between January 2018 and February 2020 are associated with the likelihood of being PCR-tested for SARS-CoV-2 and testing positive between 1 March 2020 and 31 August 2021. Findings suggest that reporting to eat breakfast, weekly sugary snack consumption (both low and high), participating in more out of school clubs, being able to ride a bike and being able to swim 25 metres were associated with an increased likelihood of being tested for SARS-CoV-2. Health behaviours associated with an increased likelihood of testing positive for SARS-CoV-2 were eating breakfast, engaging in higher weekly physical activity, participating in more out of school clubs and riding a bike. Boys were more likely to test positive for SARS-CoV-2 than girls, and those living in a less deprived area less likely to test positive than those residing in the most deprived area. (page 18 line 513-523)

16) Could you discuss whether you think the timing of the exposure measurement is a relevant time period? Would you expect sleep duration (or any of the exposures) up to 6 years before the pandemic to be associated with COVID-19 risk?

This is a crucial point that you have raised and something we have reflected on in detail. In our original submission, we wanted to examine historical health behaviours reported during primary school and subsequent testing during the adolescent period. However upon reflection, we agree that this time window may not be relevant for COVID-19 risk up to 6 years later. To address your concerns regarding timing of the exposure measurement, we decided to significantly narrow this time window to 1 January 2018 to 29 February 2020 and capture “pre-pandemic” health behaviours in line with your previous suggestion. We also removed the adolescent age criteria to examine pre-pandemic behaviours and testing/testing positive for the whole sample.

As such, the revised manuscript now presents updated analyses of multivariable regression (including binary, ordinal and continuous responses to retain power and nuance in our estimates) of pre-pandemic health behaviours between 1 January 2018 to 28 February 2020 and association with being tested for SARS-CoV-2 or testing positive for SARS-CoV-2 between 1 March 2020 and 31 August 2021. We believe this revision has significantly strengthened the value of the paper in capturing a more accurate reflection of pre-pandemic health behaviours. This has been amended throughout the manuscript, for example:

Background: This study investigates the association of pre-pandemic (prior to 1 March 2020) health-related behaviours self-reported by children aged 8-11 years during primary school before the COVID-19 pandemic between 1 January 2018 and 28 February 2020, with two outcomes; the odds of ever having a SARS-CoV-2 PCR test and the odds of testing positive for SARS-CoV-2 during the period of study. We aim to examine whether these self-reported markers of health-related behaviours reported pre-pandemic are associated with the likelihood of; i) ever being tested for SARS-CoV-2 and ii) ever testing positive for SARS-CoV-2 between 1 March 2020 and 31 August 2021. (page 7 line 141-148)

17) A few places in the manuscript it is mentioned that the clustering of health behaviours is important, but it doesn't appear that the clustering of behaviours was examined. Was it?

Thank you for querying the term clustering of health behaviours. We have removed reference to this term in the manuscript (for example, within the background), as we believe we were not clear in our reference to clustering. We originally meant clustering of health behaviours in relation to concurrently occurring behaviours such as better nutrition (for example 5 fruit/veg portions) and physically active behaviours, however as you have stated we have not examined the clustering of behaviours in analyses as this was not within our research question, and therefore have removed the term to avoid confusion.

Reviewer 2

1) First and foremost, I would invite the authors to take a step back and consider the evidence and theoretical underpinnings of their investigation (ie, research questions). The literature mentioned in the Introduction is rather generic and does not support directly the research questions. The authors argue that this is the first studies to investigate the associations between previous health behaviours and subsequent (positive) SARS-CoV-2 test during adolescence, however no proper explanation is provided for why would one expect such associations. The references #8 to #11 mentioned in the introduction tangentially address this, and the references #12 & #13 might not be enough to justify this approach.

Thank you for raising this, we have reflected considerably on your point and have made a number of amendments. We agree that our original rationale was lacking clarity and focus. To address this, we have included additional literature and expanded on discussions of evidence to support our theoretical underpinning of the study. Academics have advocated for health-related behaviours including physical activity and nutrition to be considered in current and future pandemic scenarios. This is due to the important role these behaviours play in contributing to good cardiometabolic health and effective immune system function. However, this research to date including current and previous infectious disease has been conducted solely on the adult population. Childhood is a critical period during which health behaviours are developed with track into adolescence and adulthood. Therefore, we wanted to examine any association these health behaviours may play in being tested for, and testing positive for SARS-CoV-2:

Background: Whilst evidence has demonstrated the negative impact of the COVID-19 pandemic on children's health-related behaviours including reduced physical activity, increased sedentary behaviour and poorer nutrition[1,9], it is unclear if this association is bidirectional, that is, whether these health behaviours are associated with risk of SARS-CoV-2 infection. Within the adult population, emerging evidence suggests a plausible relationship between pre-pandemic health risk behaviours such as physical inactivity and poor nutrition with SARS-CoV-2 infection and severity of disease[10–13], and increased risk of other infectious diseases[14]. This is attributed to the important role health behaviours play in shaping cardiometabolic health and immune system function. Indeed, research shows links to the early years including critical early developmental stages with subsequent risk of developing chronic inflammation, which is associated with non-communicable disease risk and mortality during adulthood [15]. Health behaviours such as adequate nutrient intake[16] and physical activity[17] are required for the regulation and function of the immune system.

As a result, academics have advocated for consideration to be placed on the role of these health behaviours in future endemic/pandemic scenarios [17]. However, research to date has concentrated on adults, explored single health behaviours or examined those with severe COVID-19 infection and hospitalisation[18,19]. The focus of research within the childhood population has principally been placed on clinical outcomes as opposed to lifestyle outcomes, including identifying the clinical characteristics of severe infection, the presence of comorbidities, common symptoms such as cough and clinical biomarkers[20,21]. Whilst serious COVID-19 illness in children is relatively rare, mild or asymptomatic infection is common[22]. Positive SARS-CoV-2 tests require periods of self-isolation,

impacting children's physical health and wellbeing, limiting opportunities for children to engage in health-promoting behaviours essential for optimal development such as regular physical activity[9,23]. Therefore, research examining the role of these health behaviours in a childhood population within the context of the COVID-19 pandemic is warranted. (page 5-6 line 15-41)

2) Secondly, the authors do not prepare at all the reader with regards to which 'health behaviours' are being investigated. As this term is rather generic and as it transpires later in the Methods section, the options of health behaviours is rather small and unusual to use as a predictor for research questions as these.

You have raised an important point here, and we agree that the term 'health behaviours' is somewhat generic as this can encompass a wide range of behaviours and is also dependent on life stage e.g. early years, childhood, adolescence, adulthood. To address this point, we have clarified specifically the health behaviours we refer to as those recognised by the OECD as current trends in children's health behaviours, and have referred to two OECD documents to support this, see references [4-5]. Within these documents, the OECD refer to specific childhood health behaviours that should be measured in order to better design and deliver policies that promote children's health. Namely, there are physical activity and sedentary behaviour, nutrition (including fruit and vegetable consumption and sugary snack consumption) and sleep:

Background: The Organisation for Economic Co-operation and Development (OECD) recognised current trends in children's health, highlighting typical health behaviours of school-aged children that warrant further research in order to better design policies that improve children's health outcomes[4,5]. These include nutrition-related behaviours such as fruit and vegetable intake, consumption of sugary foods and breakfast consumption, physical activity and sedentary behaviours and sleep. (page 5 line 7-12)

Methods: Quantitative analysis: Health-related behaviour measures included in multivariable analyses are recognised by the OECD as typical health behaviour trends during childhood that warrant research[4,5]. These related to the behaviours from the previous day (ate breakfast, travel actively to and/or from school, number of fruit/vegetables portions consumed, number of times teeth brushed, hours of sleep), frequency of behaviours every day the previous seven days (physically active \geq 60 minutes, sedentary/screen time \geq two hours, felt tired, ate a sugary snack), and general items including participation in number of out of school clubs, can ride a bike and can swim 25 metres. (page 12 line 287-295)

[4] OECD. 'Are children active and physically healthy?'. In: Measuring What Matters for Child Well-Being and Policies. Paris: : OECD Publishing 2021. 1–295. doi:10.1787/e82fded1-en

[5] OECD. Education in the Digital Age. In: Burns T, Gottschalk F, eds. Education in the Digital Age. OECD 2020. 1–218. doi:10.1787/1209166A-EN

3) The Methods section is well written and enough information about the initial data collection and data linkage is presented. Whereas there are no issues there, I would like to return to the first point I made above and ask the researchers to reflect on why would they expect to see associations between very specific questions that ask about frequency of health behaviours in the day (eg., eating breakfast), week before (e.g., PA) or usually (e.g., can ride a bike) and (sometime) 6years later SARS-COV-2 infection?

We have reflected on the inclusion criteria for our study and in particular, the time frame for survey

responses between 2014-2020. In our original submission, we wanted to examine historical health behaviours reported during primary school and subsequent testing during the adolescent period. However upon reflection, we agree that this time window may not be relevant for COVID-19 risk up to 6 years later. To address your concerns regarding timing of the exposure measurement, we decided to significantly narrow this time window to 1 January 2018 to 29 February 2020 and capture “pre-pandemic” health behaviours. We also removed the adolescent age criteria to examine pre-pandemic behaviours and testing/testing positive for the whole sample.

Whilst the survey captures specific health behaviours endorsed by the OECD including days physically active, nutritional habits and sedentary behaviour, these often represent wider constructs such as routine at home, parental supervision and health literacy. Furthermore, the exploratory nature of our study aimed to examine if there was an association with testing/testing positive in order to better inform further work including the wider constructs the behaviours represent (such as work looking at parental monitoring).

As such, the revised manuscript now presents updated analyses of multivariable regression (including binary, ordinal and continuous responses to retain power and nuance in our estimates) of pre-pandemic health behaviours between 1 January 2018 to 28 February 2020 and association with being tested for SARS-CoV-2 or testing positive for SARS-CoV-2 between 1 March 2020 and 31 August 2021. We believe this revision has significantly strengthened the value of the paper in capturing a more accurate reflection of pre-pandemic health behaviours. This has been amended throughout the manuscript, for example:

Background: This study investigates the association of pre-pandemic (prior to 1 March 2020) health-related behaviours self-reported by children aged 8-11 years during primary school before the COVID-19 pandemic between 1 January 2018 and 28 February 2020, with two outcomes; the odds of ever having a SARS-CoV-2 PCR test and the odds of testing positive for SARS-CoV-2 during the period of study. We aim to examine whether these self-reported markers of health-related behaviours reported pre-pandemic are associated with the likelihood of; i) ever being tested for SARS-CoV-2 and ii) ever testing positive for SARS-CoV-2 between 1 March 2020 and 31 August 2021. (page 7 line 141-148)

4) Although this could be seen as an exploratory investigation, both models have an extremely poor goodness-of-fit (explained variance) (i.e., $R^2=0.006$, respectively $R^2=0.02$). The authors mentioned this aspect briefly in the results but do not expand on this in their discussion. Moving beyond the statistical significance of their results (ie, identified predictors) what is the ecological significance of these results given that over 99% of model is explained by other factors? I would be reluctant

We thank the reviewer for their considerable attention to our statistical results. We recognise that the R^2 is low, around 2%, however we are hesitant to overlook our analysis on the guidance of a single metric. Moreover, given this is a logistic regression the R^2 is pseudo, and is recognised by some to be the least important measure of ‘goodness of fit’ (DeMaris 2002). The important aspect in our research is that we focused on the theoretical plausibility of our model, i.e., what behaviours could be predictive of COVID-19 given behaviours are largely guided by parents for children of these ages. Furthermore, in terms of the ecological significance, we are not looking to make generalisable recommendations here, but rather exploring the plausibility that health behaviours are associated with COVID-19 testing and subsequent infection. We encourage other research to explore these behaviours with larger samples to draw more robust conclusions. As a last point, we are very aware that research which finds small findings, or no associations, is equally valid as studies with clear associations given the considerable publishing bias that exists in academic work.

5) The results show that positive health behaviours such as reporting the recommended level of sleep (at least nine hours), participating in at least three out of school clubs, being able to ride a bike and being able to swim 25 metres were associated with an increased likelihood of being tested for SARS-CoV-2. The authors hypothesises that this could be due to the parental involvement and health literacy but fail to suggest/consider any alternative explanations.

We agree that our original submission lacked alternative explanations of the findings in our study. To address this point, we have provided an additional explanation regarding social contacts and exposures. This has been expanded on in greater detail in the discussion, and also highlighted as a key point within the conclusion.

We have discussed this regarding the findings of physically active behaviours (e.g. through co-participation with peers, potentially increasing exposure), the findings of sex differences (e.g. through gender-typed activities with girls more likely to socialise indoors and in the presence of supervising adults), and regarding deprivation-related exposures (e.g. via increased engagement in daily activities that increase social contacts such as using public transport and inability to work from home). This has also been discussed using relevant literature, see references [39-44].

Discussion: While evidence recognises the importance of adequate nutrition[16] and physical activity[17] for cardiometabolic health and immune system function, the findings in the current study draw attention to another potential mechanism of increased contacts and exposure to SARS-CoV-2. Engagement in physically active behaviours such as out of school clubs, higher frequency of physically active days in a week and riding a bike may increase the number of social contacts of the child. Indeed, there is a wealth of evidence demonstrating that childhood physical activity participation is highly influenced by their social environment and co-participation with peers[39]. It is therefore possible that physically active children had increased social contacts and exposure to SARS-CoV-2 through co-participation of activity and play opportunities.

However, it is important to note that physical activity is an essential health behaviour required for optimal development and a range of health and wellbeing outcomes, and these findings must be considered in balance with the importance of encouraging these behaviours and providing physically active opportunities during childhood. This viewpoint was also reflected in Government guidance and risk assessments during the COVID-19 pandemic through the reopening of children's playgrounds and outdoor play spaces, with explicit reference to outdoor play and physical activity as fundamental for children's development and wellbeing[40].

Contact patterns may also explain sex differences observed in this study, as we found girls are more likely to test positive for SARS-CoV-2. In addition to age assortative mixing patterns of children, there is a developmental tendency by children to socially interact with members of the same sex and engage in gender-typed activity[41]. For girls, the location of play preferences are more likely to be indoors and in contact with supervising adults, where exposure to SARS-CoV-2 is possibly greater[42]. The findings of association between increasing age and likelihood of testing positive for SARS-CoV-2 in this study are supported by wider literature which suggests increasing susceptibility of infection in the adolescent age group compared to younger than 10 to 14 years[43].

Our findings also show an area-level social gradient. Those living in the least deprived WIMD quintiles 4 and 5 were less likely to test positive for SARS-CoV-2 compared to the most deprived quintile, which may reflect deprivation-related exposure patterns to SARS-CoV-2. Indeed, research conducted using the WIMD and English area-level deprivation indicators found adults living in the most deprived areas demonstrated differential exposures to SARS-CoV-2[44]. This included patterns of public activities such as attending work or education outside of the household, using public transport and car sharing with non-household members. This, and considerations of the deprivation-related disparities in the built environment including access to open spaces highlights the inequalities that persist in risk

of SARS-CoV-2 infection. Furthermore, whilst it is likely that children mix with others from similar demographic areas, the finding in our study may also reflect community prevalence which was not captured. (page 20-21 line 619-675)

Conclusion: Furthermore, co-participation in physically active behaviours with peers may increase exposure to SARS-CoV-2. This must be considered from a risk versus benefit approach in relation to the importance of physically active behaviours for children’s development and wellbeing. (page 21-22 line 720-818)

6) As a general observation the Introduction and Discussion are completely disconnected with each other. The former focuses extensively on parental health literacy aspect which is not investigated in this study and that is only suggested as a possible mechanism behind some poorly explaining regression models. Also the authors introduce more confusion in the Discussion by bringing in parental vaccination decisions and linking it with their suggested explanations for the findings. I would advise against adding these speculations.

We recognise that our original submission required a greater connection between the background and discussion and as such, have

We have also included additional points within the background about the parental influence of children’s health behaviours:

Background: The establishment of these health behaviours during childhood are highly influenced by parental mechanisms and monitoring behaviours, particularly in children aged under 12 [6–8]. (page 5 line 12-14)

Background: Identifying the pre-pandemic health-related behavioural characteristics of children requiring a SARS-CoV-2 test or testing positive for SARS-CoV-2 infection and hypothesising potential mechanisms through which these may operate, including exposures, socio-demographic and parental influences could yield insight to inform the current COVID-19 pandemic and future pandemic/endemic scenarios. (page 6 line 42-46)

Furthermore, we agree that the original points relating to parental vaccination do not warrant discussion in relation to our findings, so we have removed these entirely.

VERSION 2 – REVIEW

REVIEWER	Amorim, Gustavo VUMC
REVIEW RETURNED	21-Jun-2022

GENERAL COMMENTS	The authors mention risk of testing positive several times when it is the odds of having a positive test result, as I believe a logistic regression was used. line 160: was a mixed-effects model used to account for school clustering? Or only a sandwich estimator for the variance was used? Did some children have more than 1 event, e.g., tested positive more than once? If so, how many and how was it handled in the analysis?
---

	What was the reasoning to construct the multivariable models? The work uses secondary data and, as such, may offer limited information for the problem. Most of the conclusions are based on variables that were not directly observed, like parent involvement or contact partners, and are based on somewhat strong or unsupported assumptions. Perhaps, a different approach like latent class regression would be preferable. Alternatively, one could fit a separate regression for each exposure, adjusting for confounders related to that exposure, instead of fitting one large model with all variables included. I did not understand the numbers in Table Online supplemental appendix 5. How come the number of children who "tested positive" or had "no evidence of positive SARS-Cov2 test" (columns 5-6, row 1: 569 + 6,498) is higher than the number of children "tested" and "not tested SARS-Cov2" (columns 3-4, row 1: 2,764 + 4,298)? I think the 'no evidence of positive SARS-Cov2' could be further clarified. What is meant by "evidence"? Because if I understood it correctly (and I apologize otherwise), some patients were tested because they were close contacts, despite not presenting any symptoms. So, if they were not tested, they would have entered the last group "no evidence of positive SARS-Cov2". The comparisons are thus actually between 'positive vs negative/not tested'. A better description of the data would be, in my opinion, 'positive vs negative vs not tested (unknown)' or simply 'positive vs negative', but with a total sample size of 2,764 for the latter. Complete case analysis may lead to biased estimates, so any conclusions should be made very carefully. This cannot be over stressed and could be clearly stated as a limitation. Line 196: Were all 2,764 patients used in the analysis? Because from my understanding, the authors did a complete case analysis, and it seems that some patients had missing information and were, therefore, excluded. For example, in Table 1 5.7% of the data had missing information on WIMD quintiles. Were they excluded? What was the actual sample size, after excluding missing observations, used in the multivariable regressions? I may have missed it, but it would be interesting to see what the median (as well as first and third quartiles) time from survey to test was. This would provide some additional information for the reader and give some meaning from the time the survey took place to the actual event. In Table 1 (and Table 5, supplementary material) the order is inverted: the header says "n (%)", but table is "(%) n". I suggest replacing p-values = 0.000 for p.value < 0.001; p-values cannot be equal to 0. What does the reference level mean for the variables "Number of fruit/vegetables portions" and "Number of times teeth brushed"? Weren't these variables treated as continuous in the model?
--	---

VERSION 2 – AUTHOR RESPONSE

Reviewer 3:

1. The authors mention risk of testing positive several times when it is the odds of having a positive test result, as I believe a logistic regression was used.

Thank you for raising this point. We have amended this as suggested in the manuscript when referring to the logistic regression results:

Page 5 line 17-18: That is, whether these health behaviours are associated with likelihood of SARS-CoV-2 infection.

Page 20-21 line 341-344: This theory may well transcend into other behaviours, including limits and freedom in socialising with others, placing a greater likelihood of infection of illness – including COVID-19.

Page 22 line 381-383: This, and considerations of the deprivation-related disparities in the built environment including access to open spaces highlight the inequalities that persist in SARS-CoV-2 infection.

When discussing the results of the logistic regression, we have only used the term odds or likelihood.

Furthermore, when we refer to the ‘risk versus benefit’ approach within the conclusion, we are referring metaphorically to health-related behaviours, as opposed to the incorrect term risk when discussing results from logistic regression. For example the risk of discouraging physically active behaviours given findings relating to increased likelihood of a positive test, versus the benefit of encouraging physically active behaviours during childhood. We have further clarified this in the conclusion as below:

Page 3 abstract: A risk-versus-benefit approach must be considered in relation to promoting these health behaviours, given the importance of health-related behaviours such as physical activity during childhood for development.

Page 22-23 line 391-395: This must be considered from a risk-versus-benefit approach in relation to promoting these health behaviours, given the importance of health-related behaviours such as physical activity during childhood for development and wellbeing.

2. Line 160: was a mixed-effects model used to account for school clustering? Or only a sandwich estimator for the variance was used?

We used a sandwich estimator for the variance, by using the following cluster command in Stata software to account for school-level differences: `cluster (schoolcode) where schoolcode = unique ID number for different participating schools`. To address this point, we have further clarified this in the manuscript to make this clear for the reader, as presented below.

Page 12 line 178-183: Complete case multivariable logistic regression analyses, adjusting for confounding variables (sex, age on 1 March 2020, area-level deprivation using the Welsh Index of Multiple Deprivation (WIMD)[37] (version 2019) and clustered by school (using sandwich estimator to account for differences between schools), determined Odds Ratios (OR) for i) ever being PCR-tested for SARS-CoV-2 virus and ii) ever having a positive PCR SARS-CoV-2 test during the period of interest.

3. Did some children have more than 1 event, e.g., tested positive more than once? If so, how many and how was it handled in the analysis?

It is possible that some children have tested positive for SARS-CoV-2 more than once, but we were only interested in ever testing for, or ever testing positive for SARS-CoV-2 regardless during the period of interest (1 March 2020 and 31 August 2021). We do not have the figures of multiple test occurrences as this was not part of our research question or in developing our dataset. We used a flag to identify whether they had ever tested for or ever tested positive for SARS-CoV-2 during the period of interest. A binary code was assigned for ever PCR tested during period of interest (where 1==any PCR test between 1 March 2020 and 31 August 2021, 0=no PCR SARS-CoV-2 test) and ever PCR tested positive during period of interest (where 1==any positive SARS-CoV-2 test, 0=negative PCR test, 0=no PCR test).

To address this, we have stated this more explicitly in the manuscript.

Page 7 line 60-67: This study investigates the association of pre-pandemic (prior to 1 March 2020) health-related behaviours self-reported by children aged 8-11 years during primary school before the COVID-19 pandemic between 1 January 2018 and 28 February 2020, with two outcomes; the odds of ever having a SARS-CoV-2 PCR test and the odds of ever testing positive for SARS-CoV-2 during the period of study. We aim to examine whether these self-reported markers of health-related behaviours reported pre-pandemic are associated with the likelihood of; i) ever being tested for SARS-CoV-2 and ii) ever testing positive for SARS-CoV-2 between 1 March 2020 and 31 August 2021.

Page 11 line 163-170: The primary outcomes were i) whether the child was ever PCR-tested for the SARS-CoV-2 virus and ii) whether the child had any positive SARS-CoV-2 test between 1 March 2020 and 31 August 2021. Participants were assigned a binary code for any SARS-CoV-2 test during the period of interest (1: PCR-tested at least once for SARS-CoV-2 between 1 March 2020 and 31 August 2021; 0: no PCR SARS-CoV-2 test) and again for any positive SARS-CoV-2 test during period of interest (1: any positive SARS-CoV-2 PCR test between 1 March 2020 and 31 August 2021; 0: negative PCR test for SARS-CoV-2; 0: not PCR-tested for SARS-CoV-2 (unknown)). In the case of multiple PCR tests, the first occurrence was used.

Page 18 line 296-299: This study examines whether markers of health-related behaviours reported by primary school-aged children between January 2018 and February 2020 are associated with the likelihood of ever being PCR-tested for SARS-CoV-2 and ever testing positive between 1 March 2020 and 31 August 2021.

4. What was the reasoning to construct the multivariable models?

The first submission of this manuscript presented a manual step-wise approach to model building. Comments from the first round of peer review included further clarity regarding the model building approach and suggested further theoretical underpinning. There, we consulted with EL who provided statistical guidance and suggested that an exploratory theory-based model may provide better insight. From this, we changed our step-wise approach and explored all areas of health behaviours concurrently. We recognise that our previous modelling strategy may have been weaker given our unclear rationale. In our revisions, we examined all health behaviours in a univariate model, then all together in an unadjusted multivariate model (online supplemental appendix 5), and then we adjusted our multivariate model for confounders (presented in Tables 2 and 3, pages 15-18).

Within the resubmission, we included additional literature and strengthened our rationale for including these health behaviours in a multivariable model. Previous research demonstrates the importance of these health behaviours for cardiometabolic health and immune system function, and academics have advocated for greater importance to be placed on these health behaviours in future pandemics. This supports the exploratory theory-based model as suggested by EL. Furthermore, the health behaviours explored within analyses are recognised by the OECD as current trends in children's health behaviours, refer to reference [4-5], and we clarified further in the background and methods, for example:

Page 5-6 line 19-41: Within the adult population, emerging evidence suggests a plausible relationship between pre-pandemic health risk behaviours such as physical inactivity and poor nutrition with SARS-CoV-2 infection and severity of disease[10–13], and increased risk of other infectious diseases[14]. This is attributed to the important role health behaviours play in shaping cardiometabolic

health and immune system function. Indeed, research shows links to the early years including critical early developmental stages with subsequent risk of developing chronic inflammation, which is associated with non-communicable disease risk and mortality during adulthood[15]. Health behaviours such as adequate nutrient intake[16] and physical activity[17] are required for the regulation and function of the immune system.

As a result, researchers have advocated for consideration to be placed on the role of these health behaviours in future endemic/pandemic scenarios[17]. However, research to date has concentrated on adults, explored single health behaviours or examined those with severe COVID-19 infection and hospitalisation[18,19]. The focus of research within the childhood population has principally been placed on clinical outcomes as opposed to lifestyle outcomes, including identifying the clinical characteristics of severe infection, the presence of comorbidities, common symptoms such as a cough and clinical biomarkers[20,21]. Whilst serious COVID-19 illness in children is relatively rare, mild or asymptomatic infection is common[22]. Positive SARS-CoV-2 tests require periods of self-isolation, impacting children's physical health and wellbeing, limiting opportunities for children to engage in health-promoting behaviours essential for optimal development such as regular physical activity[9,23]. Therefore, research examining the role of these health behaviours in a childhood population within the context of the COVID-19 pandemic is warranted.

5. The work uses secondary data and, as such, may offer limited information for the problem. Most of the conclusions are based on variables that were not directly observed, like parent involvement or contact partners, and are based on somewhat strong or unsupported assumptions. Perhaps, a different approach like latent class regression would be preferable. Alternatively, one could fit a separate regression for each exposure, adjusting for confounders related to that exposure, instead of fitting one large model with all variables included.

Thank you for raising this point. Parents have considerable control and decision-making over child behaviours, and it is not unusual for child health behaviours to often represent underlying parental autonomy. As we were interested in each variables contribution to COVID-19, a latent class analysis would not be preferable in this instance, but it is an interesting idea for another paper to potentially explore health behaviour clustering and agree each approach has its strengths. In the original paper we conducted more of a step-wise approach but both previous reviewers were not accepting of this. We have since changed the analysis to become more exploratory due to the new nature of the analysis, hence why we have included many covariates which we theoretically believe construct health behaviours in children, and then adjusted these for relevant confounders.

Furthermore, we have included important literature within the background and discussion that spans the fields of parental influence and involvement, health literacy and association with child health behaviours and outcomes, and feel this evidence supports our theory. For example, see refs: [6-8], [39-43]. Finally, we clarify this as theory based on the evidence Through this, we theorise that because health behaviours are largely guided and facilitated by parents, our associations are likely to be reflecting health literacy among parents, along with monitoring behaviours (page 19 line 311-313).

6. I did not understand the numbers in Table Online supplemental appendix 5. How come the number of children who "tested positive" or had "no evidence of positive SARS-Cov2 test" (columns 5-6, row 1: 569 + 6,498) is higher than the number of children "tested" and "not tested SARS-Cov2" (columns 3-4, row 1: 2,764 + 4,298)?

Thank you for drawing this to our attention, we have checked the data and the figure relating to no evidence of positive test (6,498) was a mistype. This has been amended to the correct figure, 6,493 in Table 1 (13-14) and in online supplemental appendix 5.

7. I think the 'no evidence of positive SARS-Cov2' could be further clarified. What is meant by "evidence"? Because if I understood it correctly (and I apologize otherwise), some patients were tested because they were close contacts, despite not presenting any symptoms. So, if they were not tested, they would have entered the last group "no evidence of positive SARS-Cov2". The comparisons are thus actually between 'positive vs negative/not tested'. A better description of the data would be, in my opinion, 'positive vs negative vs not tested (unknown)' or simply 'positive vs negative', but with a total sample size of 2,764 for the latter.

Thank you for raising this point. We agree that the term no evidence of positive SARS-CoV-2 test could be further clarified for the reader. We originally used the term evidence in relation to the presence of PCR testing data, and wanted to avoid the term negative as our definition of no evidence of positive SARS-CoV-2 test includes absence of any PCR testing data (i.e. not just a negative result)

The binary coding within our analyses (outlined on page 11, Quantitative analysis sub section, lines 165-169) is as follows:

Participants were assigned a binary code for any SARS-CoV-2 test during the period of interest (1: PCR-tested at least once for SARS-CoV-2 between 1 March 2020 and 31 August 2021; 0: no PCR SARS-CoV-2 test) and again for any positive SARS-CoV-2 test during period of interest (1: any positive SARS-CoV-2 PCR test between 1 March 2020 and 31 August 2021; 0: negative PCR test for SARS-CoV-2; 0: not PCR-tested for SARS-CoV-2 (unknown)).

We appreciate your suggestion and agree the term suggested would provide more clarity. To address your point, we have amended this as suggested to Tested negative/not tested (unknown) for SARS-CoV-2 in Table 3 and online supplemental appendix 5, for example see below:

	Tested for SARS-CoV-2	Not tested for SARS-CoV-2	Tested positive for SARS-CoV-2	Tested negative/not tested (unknown) for SARS-CoV-2
	% (n)	% (n)	% (n)	% (n)

We have also removed the term ‘evidence of’ in relation to testing within the body of the manuscript. For example page 11 line 154-155 (1; any positive SARS-CoV-2 PCR test between 1 March 2020 and 31 August 2021, 0; negative PCR test for SARS-CoV-2, 0; not PCR tested for SARS-CoV-2 (unknown)).

8. Complete case analysis may lead to biased estimates, so any conclusions should be made very carefully. This cannot be over stressed and could be clearly stated as a limitation.

Thank you for raising this important point. In addition to our response below, we have provided further detail in response to point 9 below.

We have addressed this further within the manuscript, and within the strengths and limitations subsection adhering to BMJ Open journal requirements and within the limits of previous editorial requests during the first round of revisions which included “This section should contain up to five short bullet points, no longer than one sentence each, that relate specifically to the methods. The novelty, aims, results or expected impact of the study should not be summarised here”

Page 3 strengths and limitations:

Reporting of multiple child health behaviours linked at an individual-level to routine records of SARS-CoV-2 testing data through the SAIL Databank, using complete case analysis.

Page 11-12 line 174-187: Complete case multivariable logistic regression analyses, adjusting for confounding variables (sex, age on 1 March 2020, area-level deprivation using the Welsh Index of Multiple Deprivation (WIMD)[37] (version 2019) and clustered by school (using sandwich estimator to account for differences between schools), determined Odds Ratios (OR) for i) ever being PCR-tested for SARS-CoV-2 virus and ii) ever having a positive PCR SARS-CoV-2 test during the period of interest. Missing categories of data (sex and WIMD data obtained through the SAIL Databank) were tested to see if they significantly predicted any outcomes.

Page 13 line 227-234: We tested if missing categories of data (sex and WIMD obtained through the SAIL Databank) significantly predicted any outcomes and found that no missing categories significantly predicted the outcomes. Therefore, missing data were assumed to be at random through data linkage[38]

Dong Y, Peng CYJ. Principled missing data methods for researchers. Springerplus 2013;2:1–17. doi:10.1186/2193-1801-2-222

Table 1 (page 15-16) footnote: OR: Odds Ratio; 95% CI: 95% confidence intervals; ** $p < 0.05$, * $p < 0.1$. See online supplemental appendix 4 for variable codebook. Low to moderate correlation between variables (coefficients -0.19 to 0.71). Complete case analysis.

Table 2 (page 17-18) footnote: OR: Odds Ratio; 95% CI: 95% confidence intervals; ** $p < 0.05$, * $p < 0.1$. See online supplemental appendix 4 for variable codebook. Low to moderate correlation between variables (coefficients -0.19 to 0.71). Complete case analysis.

9. Line 196: Were all 2,764 patients used in the analysis? Because from my understanding, the authors did a complete case analysis, and it seems that some patients had missing information and were, therefore, excluded. For example, in Table 1 5.7% of the data had missing information on WIMD quintiles. Were they excluded? What was the actual sample size, after excluding missing observations, used in the multivariable regressions?

Further to the points above, we tested if missing categories of data significantly predicted any outcomes, and we found that no missing categories significantly predicted the outcomes. As our models only have 7% missing data (maximum), and this is likely due to data linkage errors. For example the largest proportion of missing data (7% maximum) is from data obtained through data linkage of sex and WIMD deprivation data. Missing data from child-reported health behaviours through the HAPPEN survey are minimal (e.g. only 2 out of 13 survey variables contain missing any responses, maximum 1.7%, see online supplemental appendix 5). Therefore, we deem our missing data to not be a cause for concern in terms of biasing the estimates.

This is supported by evidence (below), where the authors state: “The proportion of missing data is directly related to the quality of statistical inferences. Yet, there is no established cutoff from the literature regarding an acceptable percentage of missing data in a data set for valid statistical inferences. For example, Schafer (1999) asserted that a missing rate of 5% or less is inconsequential. Bennett (2001) maintained that statistical analysis is likely to be biased when more than 10% of data are missing. Furthermore, the amount of missing data is not the sole criterion by which a researcher assesses the missing data problem. Tabachnick and Fidell (2012) posited that the missing data mechanisms and the missing data patterns have greater impact on research results than does the proportion of missing data”

Dong, Y., & Peng, C. Y. (2013). Principled missing data methods for researchers. SpringerPlus, 2(1), 222. <https://doi.org/10.1186/2193-1801-2-222>

Please find below further amendments in text below, and the addition of Dong and Peng’s (2013) paper as an additional reference (see ref 38).

Page 12 line 185-187: Missing categories of data (sex and WIMD data obtained through the SAIL Databank) were tested to see if they significantly predicted any outcomes.

Page 13-14 line 226-234: The maximum missing data was 7% (see Table 1). We tested if missing categories of data (sex and WIMD obtained through the SAIL Databank) significantly predicted any outcomes and found that no missing categories significantly predicted the outcomes. Therefore, missing data were assumed to be at random through data linkage[38].

10. I may have missed it, but it would be interesting to see what the median (as well as first and third quartiles) time from survey to test was. This would provide some additional information for the reader and give some meaning from the time the survey took place to the actual event.

Thank you for this useful suggestion, we agree this would provide additional information for the reader regarding time between survey participation and date of test/positive SARS-CoV-2 test.

To address this, we have added this information to Table 1 and online supplemental appendix 5 (full descriptive statistics table), in addition to a short description in the text:

Page 13 line 223-226: The time between the HAPPEN survey date and SARS-CoV-2 PCR test date (median number of days (interquartile range)) was 588 (385 – 685) days for being PCR-tested and 672 (599 – 715) days for PCR testing positive for SARS-CoV-2.

Table 1:

	Tested for SARS-CoV-2 % (n)	Not tested for SARS-CoV-2 % (n)	Tested positive for SARS-CoV-2 % (n)	Tested negative/not tested (unknown) for SARS-CoV- 2 % (n)
Sample	39.1% (2,764)	60.9% (4,298)	8.1% (569)	91.9% (6,493)

Age at time of HAPPEN survey	10.1 ± 0.8	9.9 ± 0.9	10.1 ± 0.8	9.9 ± 0.8
Age on 01/03/2020 (start of period of interest)	10.6 ± 0.9	10.3 ± 1.1	10.6 ± 1.0	10.4 ± 1.0
Number of days between HAPPEN survey and SARS-CoV-2 test (median (IQR))	588 (385 – 685)		672 (599 – 715)	

11. In Table 1 (and Table 5, supplementary material) the order is inverted: the header says "n (%)", but table is "(%) n".

Thank you for drawing this to our attention, we have corrected this oversight in Table 1 and Table 5, and as below:

Table 1

	Tested for SARS-CoV-2 % (n)	Not tested for SARS-CoV-2 % (n)	Tested positive for SARS-CoV-2 % (n)	No evidence of positive SARS-CoV-2 test % (n)
--	--------------------------------	------------------------------------	---	--

12. I suggest replacing p-values = 0.000 for p.value < 0.001; p-values cannot be equal to 0.

Thank you for this suggestion, we have amended this in the manuscript as suggested and in the online supplemental appendix 6.

Table 2 page 15:

Age 01/03/2020	1.25**	< 0.001	1.16 – 1.35
----------------	--------	---------	-------------

Online supplemental appendix 6:

Can swim 25m	1.30**	< 0.001	1.15 – 1.46
Reference: cannot swim 25m	1.00		

13. What does the reference level mean for the variables "Number of fruit/vegetables portions" and "Number of times teeth brushed"? Weren't these variables treated as continuous in the model?

Thank you for querying this point and agree that the reference level for these two variables is stated incorrectly as they are continuous variables. We have removed this in Tables 2 and 3.

Page 12 line 198-199: A list of variables included in analyses, coding response categories and coding framework is presented in online supplemental appendix 4.

Footnote of Table 2 and 3: See online supplemental appendix 4 for variable codebook.

And finally, we have made a number of minor changes throughout the manuscript to correct typographic errors or to improve the narrative/readability.

VERSION 3 – REVIEW

REVIEWER	Amorim, Gustavo VUMC
REVIEW RETURNED	27-Jul-2022
GENERAL COMMENTS	I would like to thank the authors for addressing all my questions. I have no further comments.